# Harnessing screw dislocations in shell-lattice metamaterials for efficient, stable electrocatalysts

Liqiang Wang[1,9], Di Yin[2,9], James Utama Surjadi [3,9], Junhao Ding [4], Huangliu Fu[5], Xin Zhou[6], Rui Li[4], Mengxue Chen[7], Xinxin Li[8], Xu Song [4] ✉, Johnny C. Ho [2] ✉ & Yang Lu [8] ✉

Developing highly active and robust catalysts remains a critical challenge for the industrial realization and implementation of nitrate reduction. Here, we proposed a screw dislocation-mediated three-dimensional (3D) printing strategy for scalable, integrated manufacturing of metamaterial catalysts. Specifically, screw dislocation was introduced into the 3D printing process to mediate the simultaneous synthesis of 3D architecture and chiral surface nanostructures, effectively eliminating conventional heterointerfaces. Additionally, severe strain effects induced by dislocation multiplication in curved spaces enhance intrinsic catalytic activity by promoting $NO_3^-$ adsorption and lowering the energy barrier of $NO_3^-$-to-$NH_3$ conversion. Consequently, the FeCoNi dual-scale shell-lattice metamaterials with high dislocation density achieve a Faraday efficiency of 95.4%, an $NH_3$ yield rate of 20.58 mg h$^{-1}$ cm$^{-2}$, and long-term stability exceeding 500 hours. A flow-through electrolyzer coupled with an acid absorption unit successfully produced $NH_4Cl$ fertilizer products. Our work opens a new perspective for advancing 3D printing technology in catalysis applications.

Electrocatalysis is an emerging redox platform that can achieve sustainable green energy conversion and wastewater treatment under mild conditions. Electrochemical nitrate reduction reaction ($NO_3RR$) represents a promising route for enabling $NO_3^-$ removal and value-added ammonia ($NH_3$) synthesis. However, electrocatalysts used in industrial-scale $NO_3RR$ are often subjected to harsh conditions, such as large current density, elevated temperature, and mechanical stresses, which bring challenges to the mechanical stability of electrocatalysts[1–3]. The mechanical stability of an electrode is mostly determined by the electrocatalyst-support interfaces. For most catalytic studies, catalytically active nanomaterials are typically wash-coated onto porous inert catalyst supports using polymer binders such as Nafion[1]. Thus, the formation of electrocatalyst-substrate heterointerfaces with remarkably distinct physical properties may trigger a series of interface issues during prolonged use, such as low mechanical stability[4], serious Joule heating from high contact resistance[5], and chemical dissolution in the interfacial region[6]. The development of binder-free self-supporting catalysts[1,7–9], defined by their direct growth on the support with the electrostatic adsorption or intermolecular attraction induced by van der Waals force[10,11], partially improved the

[1]Department of Mechanical Engineering, City University of Hong Kong, Hong Kong, China. [2]Department of Materials Science and Engineering, City University of Hong Kong, Kowloon, China. [3]Department of Mechanical Engineering, Massachusetts Institute of Technology, Cambridge, MA, USA. [4]Department of Mechanical and Automation Engineering, The Chinese University of Hong Kong, Hong Kong, China. [5]Shenyang National Laboratory for Materials Science, Institute of Metal Research, Chinese Academy of Sciences, Shenyang, China. [6]Physikalisches Institut, Westfälische Wilhelms-Universität, Münster 48149, Germany. [7]Department of Chemistry, City University of Hong Kong, Hong Kong, China. [8]Department of Mechanical Engineering, The University of Hong Kong, Hong Kong, China. [9]These authors contributed equally: Liqiang Wang, Di Yin, James Utama Surjadi. ✉e-mail: xsong@mae.cuhk.edu.hk; johnnyho@cityu.edu.hk; ylu1@hku.hk

binding strength with the substrate. However, these catalysts still suffered from various heterointerface issues, limiting their ability to fully address the aforementioned issues. As such, a promising strategy involves integrating nanomaterial design and support fabrication into a unified process, thereby fundamentally avoiding the heterointerface.

The recent advancements in additive manufacturing (3D printing) technology allow the one-step integrated manufacturing of end-use components[12], shifting from "assembled parts" into a single "integral component", providing a new perspective to overcome the persistent electrocatalyst-substrate heterointerface issue. However, leveraging additive manufacturing technology to directly produce macroscale components with integrated nanoscale features has been challenging[13]. Existing studies on 3D-printed metamaterial catalysts have primarily depended on exposing electrochemically active sites through various surface treatment processes at the post-printing stage, including electrochemical deposition[14,15], heterogeneous growth of low-dimensional nanomaterials[16,17], and dealloying processes[18]. These multi-step approach not only compromises the integrated manufacturing advantage of 3D printing but also continues to face the interface stability challenges in conventional electrocatalysts. To achieve one-step integrated manufacturing of multiscale features within a single catalyst, it is imperative to identify the intersection between active nanomaterial growth and the additive manufacturing process.

Fundamentally, the synthesis of arbitrary nanostructures mirrors the bottom-up process of crystal growth, akin to additive manufacturing, as it involves the nucleation and epitaxial growth of crystals[19]. As a typical line defect in crystals, screw dislocations can shear part of the crystal lattice along the out-plane direction and continuously create "new layers" by propagating the self-perpetuating growth steps[20]. Screw dislocation has been simultaneously recognized as an effective strategy for facilitating the epitaxial growth of bulk crystals and enabling the formation of various novel anisotropic nanostructures[20–23], establishing itself as an outstanding candidate. Additionally, the lattice strain induced by dislocations can manipulate the electronic structure to enhance the adsorption energy of reaction intermediates on the catalyst surface[24], thus showing the potential for enhancing the intrinsic catalytic activity of electrochemical reactions.

Therefore, in this work, screw dislocation was utilized as a bridge to connect the macroscale electrode manufacturing and low-dimensional nanomaterials synthesis. We propose a scalable, integrated manufacturing strategy for metamaterial catalysts that incorporates screw dislocations into the 3D printing process, enabling the simultaneous synthesis of a three-dimensional framework and an interface-free surface nanostructure. This monolithic nature of the metamaterial catalyst prevents the peeling of active nanomaterial and provides high mechanical stability. Additionally, dislocation multiplication in curved space creates large and inhomogeneous lattice strain fields. These combined effects endow multicomponent dual-scale shell-lattice metamaterials with superior catalytic activity and long-term stability, sustaining nitrate-to-ammonia conversion reaction for up to 500 hours. Density functional theory (DFT) calculations suggest the synergistic effect of step defects and lattice strain in facilitating $NO_3^-$ adsorption and lowering the energy barrier for $NO_3^-$-to-$NH_3$ conversion.

## Results
### Screw dislocation-mediated integrated manufacturing of metamaterial catalysts
In this work, the 3D hydrogel scaffold was first fabricated via the digital light processing (DLP) technique[25], then immersed in a metal salt precursor solution containing nickel nitrate, iron nitrate, and cobalt nitrate. Subsequently, obtained metal-salt-rich hydrogel architecture was calcinated in air to form metal oxides, and then reduced in a hydrogen atmosphere to produce FeCoNi multicomponent

metamaterials with an equiatomic atomic ratio (Supplementary Fig. 1). We observed a uniform linear shrinkage at each hierarchical level of approximately 72.1%, accompanied by an estimated 92.8% mass loss of sample during the calcination process (Supplementary Fig. 2 and Supplementary Table 1). The specific roles of each fabrication step in the development of the metamaterial catalyst are elaborated in Supplementary Note 1 and Supplementary Fig. 3. A dual-scale triply periodic minimal surface (TPMS) gyroid metamaterial with non-self-intersecting periodic porous structures was employed as the model architecture, designed through a Boolean intersection operation within the implicit modeling methodology (Supplementary Note 2)[26]. At any given relative density and specific surface area, dual-scale TPMS design strategy can achieve drastically higher mean absolute curvatures compared to single-order lattices. Based on the classical Burton-Cabrera-Frank crystal theory[27], we finally tuned the supersaturation of metal salt precursor and thermal treatment parameter (Supplementary Fig. 4) to achieve the growth mode transition from layer-by-layer (LBL) uniform growth to screw dislocation-driven spiral growth. This resulted in the simultaneous synthesis of the 3D architecture and abundant surface nanostructure.

SEM images of the FeCoNi metamaterials reveal hierarchical 3D topological features spanning seven orders of magnitude in length scale (from nanometers to centimeters) (Fig. 1a and Supplementary Fig. 5). Each gyroid unit cell contains a first-order pore diameter of ~900 μm that is comprised of a second-order gyroid network with the thin wall thickness of ~10 μm. At the nanometer scale, the step-like nanostructure is uniformly distributed throughout the volume of each thin wall in the shell-lattice metamaterials. Polycrystalline grain boundaries constrained the growth propagation of screw dislocation in the in-plane direction, manifesting a diverse array of two-dimensional patterns (Fig. 1b–d and Supplementary Fig. 6). The atomic force microscopy (AFM, Fig. 1c) results clearly revealed an average step thickness of approximately 8 nm (Supplementary Fig. 7). The metamaterials exhibit a bulk scale $10^7$ times larger than their smallest nanoscale feature sizes within the metamaterials, establishing them as promising candidates for high-performance catalytic electrodes.

Interestingly, unique micrometer-sized chiral helicoids were widely observed at the edge of the thin wall (Fig. 1e–g). The polygon-shaped layer subunits are organized in a helical structure around a central axis, with each lower layer twisting and diverging from the center following the arrangement of the upper layer. This suggests that new screw dislocation sites are continuously forming on the surface of the lower layer, leading to the successive creation of new layers. The twist angles are plotted as a function of layer number from the bottom of the spiral to the top of the spiral in Fig. 1h. An approximately linear relationship between the angle and layer number was observed with an average twist of 6.1° per layer. Then, selected area electron diffraction (SEAD) in TEM was employed to confirm the twist behavior of nanosteps. Multiple sets of electron diffraction patterns along the [100] zone axis were detected simultaneously inside the same grain with misorientation angles of 5° and 10° (Supplementary Fig. 8), indicating a twist-induced lattice misfit among the nanolayers.

### Dislocation analysis in shell-lattice metamaterials
High-angle angular dark field scanning transmission electron microscopy (HAADF-STEM) was employed to observe the cross-sectional microstructure. In contrast to the dislocation-starved polycrystal structure observed in the sample with LBL uniform growth, this optimized sample exhibited large-scale dislocation networks (green line) and sharp step-like nanostructure (red line) (Fig. 2a). STEM-EDS and atomic probe tomography (APT) (Supplementary Fig. 9-10) revealed the absence of element segregation at the nanosteps position. Diffraction contrast TEM is a powerful technique for imaging dislocations in crystals, leveraging additional electron diffraction caused by the

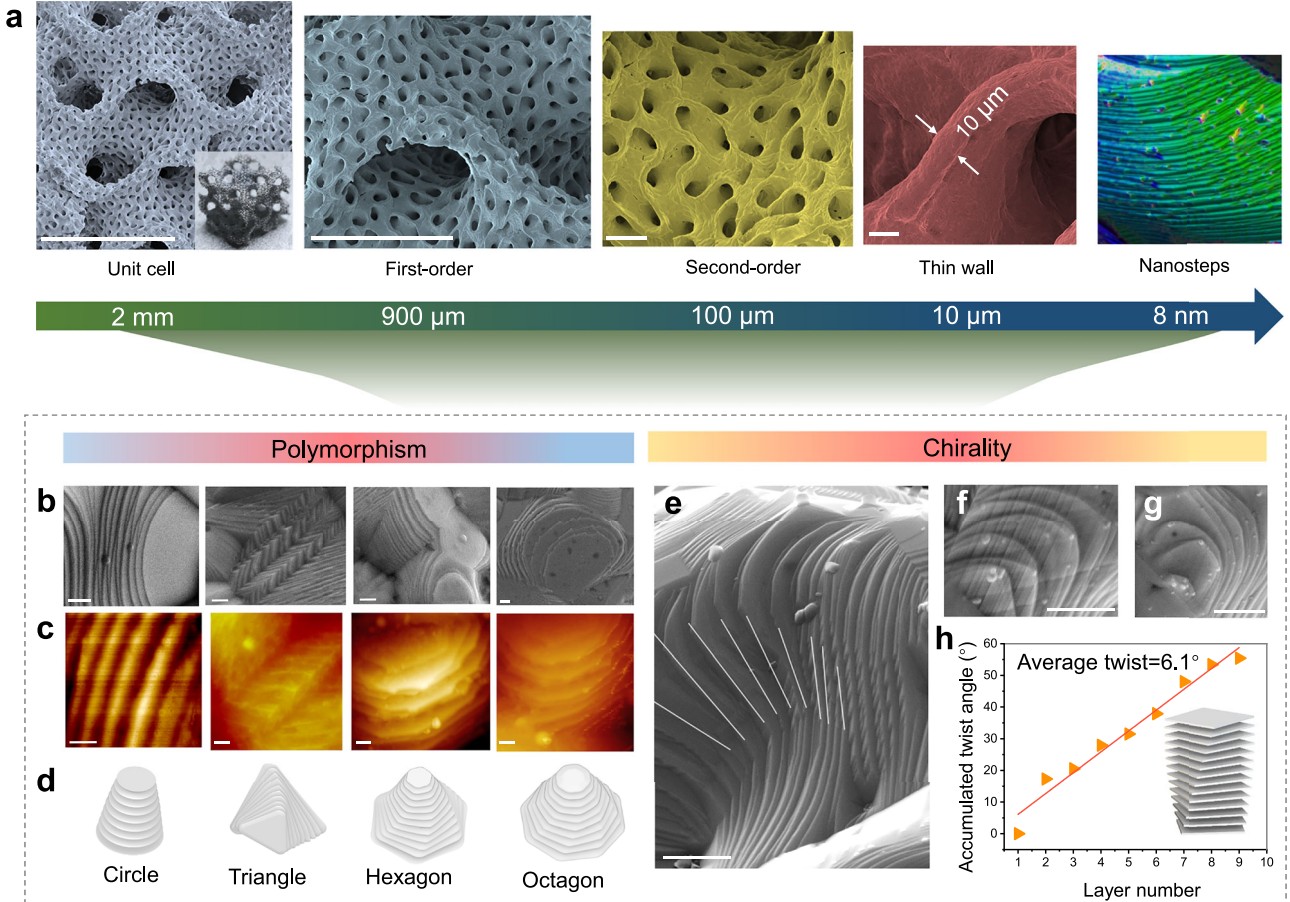

**Fig. 1 | Morphology and microstructure of integrated and scalable metamaterial catalyst. a** Scanning electron micrographs showing the breakdown of the architecture hierarchy of the shell-lattice metamaterial, including a first-order unit cell of 2 mm, a first-order pore diameter of -900 μm, a second-order pore diameter of -100 μm, smooth thin walls with a feature size down to -10 μm in wall thickness, and twisted nanosteps. Scale bar: 2 mm for unit cell, 900 μm for first-order structure, 100 μm for second-order structure, and 10 μm for thin-wall thickness. **b** SEM images depicting various nanostep morphologies. Scale bar: 500 nm. **c** AFM images displaying various zoom-in nanostep morphologies. Scale bar: 200 nm. **d** Schematic diagrams demonstrate the polymorphism varying from circles to the octagons of nanosteps morphology. **e**–**g** Representative chirality feature of nanoplates. Scale bar: 1 μm. **h** Scatter plot and linear fitting curve of the evolving twist angles θ in **e** as a function of layer number counted from the bottom to the top. The average θ is equal to 6.1°. Source data are provided as a Source Data file.

bending of atomic planes near the dislocation core. Then, we conducted a diffraction contrast TEM experiment under two-beam conditions to determine the dislocation type (Fig. 2b). The diffraction contrast of dislocation was imaged with different $g$ reflections in reciprocal space under the [110] zone axis. According to the dislocation invisibility criterion, $g * b = 0$, where $b$ is the Burgers vector and $g$ is the reciprocal lattice vectors. We observed that the $(0\bar{2}2)$ diffraction spot displayed high dislocation contrast, which is nearly perpendicular to the growth axis of step. A similar dislocation contrast can also be observed under $(\bar{1}11)$. However, when exciting the $(1\bar{1}1)$ family of spot, most of dislocations became invisible. The Burger vector was determined to be along [110] direction. The identified line direction is compared with the Burgers vector direction. If the angular difference between the Burgers vector and the dislocation line is less than 10 degrees, then the dislocation character is determined. If two characters are possible, the one with the closest match to the dislocation line is chosen. Therefore, multiple dislocation arrays in the same plane along the growth axis were confirmed to be close to pure screw dislocations (Fig. 2c). A few dislocations are observed near the edge character (Fig. 2c). Statistical analysis from corresponding STEM image further revealed that 78% of the dislocation were screw-type, while 22% were edge-type, indicating that the growth of nanosteps was governed by screw dislocation. Based on the XRD diffraction peak broadening effect, we employed the macroscale Williamson–Hall method[28] to

estimate total dislocation density of $2.23 \times 10^{14}$ m$^{-2}$ (Fig. 2d and Supplementary Fig. 11). In addition, the geometrically necessary dislocation (GND) calculated from the electron backscattered diffraction (EBSD) data (Supplementary Fig. 12) for the corresponding sample was $1.35 \times 10^{14}$ m$^{-2}$, demonstrating the reasonable measurement by XRD. Compared to previous catalysts fabricated by conventional methods, our 3D-printed metamaterial catalysts exhibited comparable or even higher dislocation density (Supplementary Table 2).

Kernel average misorientation (KAM) analysis from EBSD measurement can visualize dislocations density distribution at a millimeter scale by quantifying local lattice curvature and the distortion of crystal[29]. The 3D cross-sectional KAM map in Fig. 2e reveals high stored strain energy induced by abundant dislocation network in the porous shell-lattice metamaterials. From HRTEM images, a clear edge dislocation (Fig. 2f) with Burgers vectors $b = 1/2[01\bar{1}]$ is identified using the Burgers circuit[30]. Geometric phase analysis (GPA) results demonstrated that lattice strain fields produced by dislocations are highly inhomogeneous, with the edge dislocation inducing a pronounced linear strain field in the lattice. The maximum strain occurs at the core of the dislocation, extending its influence over a significant region of approximately 3 nm. Another closed circuit displayed a screw dislocation (Fig. 2g) with Burgers vectors $b = 1/3[111]$. The corresponding strain map shows butterfly-shaped feature, indicative of more serious lattice distortion.

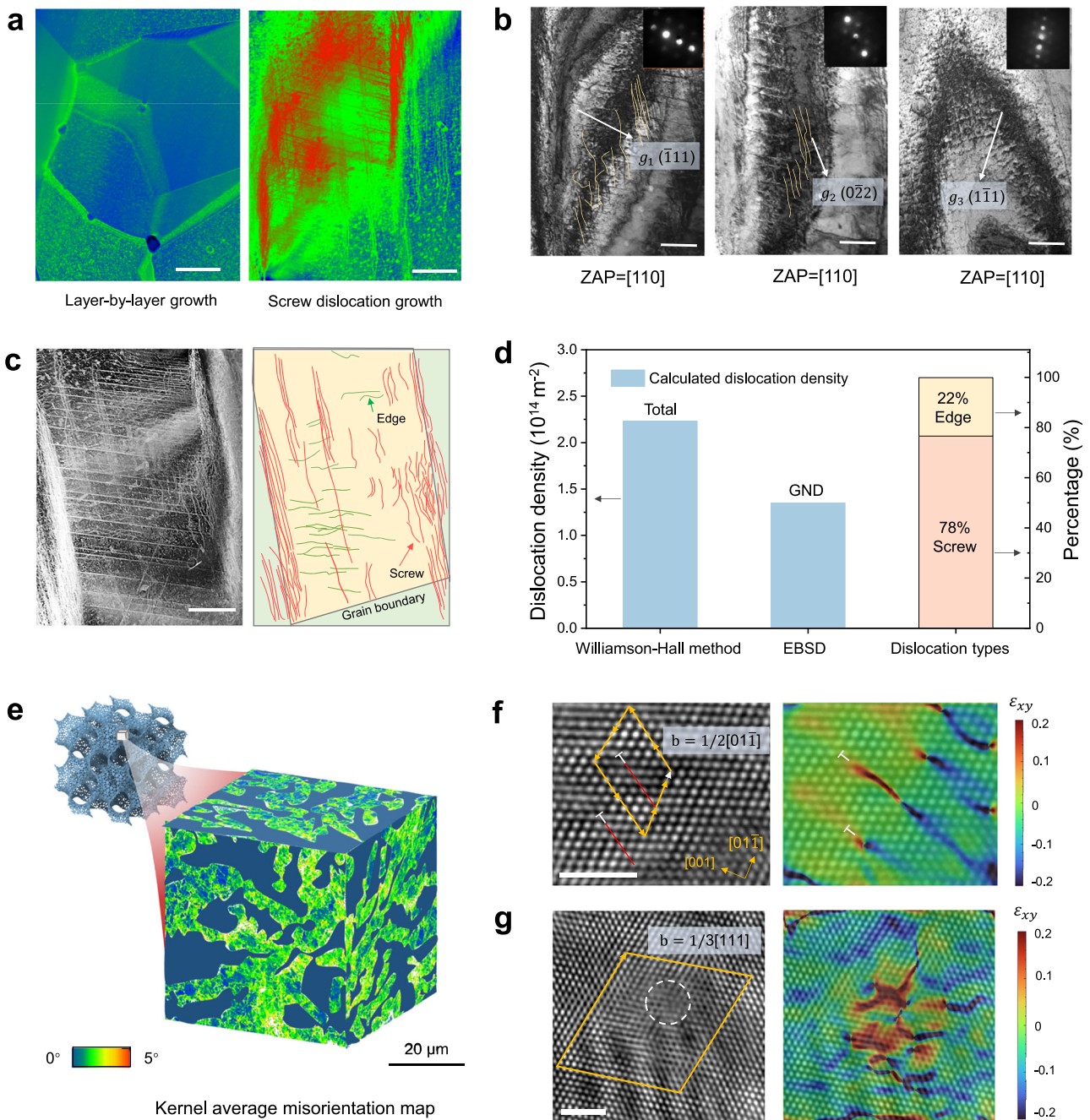

**Fig. 2 | Dislocation analysis in dual-scale shell-lattice metamaterials. a** High-angle angular dark field scanning transmission electron microscopy (HAADF-STEM) cross-section images contrasting the morphology of dislocation-starved grain structure in layer-by-layer growth mode and the dislocation-rich grain structure in screw dislocation growth mode. The sharp steps (red line) penetrated by multiple dislocation arrays (green line). Scale bar: 500 nm. **b** Diffraction contrast TEM imaging of the dislocation under two-beam conditions of [1̄10] zone axis. TEM images with selected area electron diffraction (SAED) patterns inset under strong two-beam conditions of (1̄11), (02̄2), and (11̄1), respectively. The diffraction contrast was strong when the **g** vector parallel to the dislocation direction, while became invisible when the **g** vector was excited in (11̄1). The calculated Burgers vector direction was [1̄10], suggesting the Burgers vector of this dislocation had a large screw component. Scale bar: 300 nm. **c** HAADF-STEM images and corresponding schematic present the distribution of different dislocation line types. Scale bar: 500 nm. **d** Statistical result revealing the total dislocation density, geometrically necessary dislocation (GND), and the percentage of different dislocation types. **e** Three-dimensional Kernel average misorientation (KAM) map showing the high-density dislocations in shell-lattice metamaterials. The color scale indicates the misorientation angle range. High-resolution TEM (HRTEM) image displaying the typical (**f**) edge dislocation and (**g**) screw dislocation, as well as their corresponding Burgers vector and geometric phase analysis (GPA) strain analysis. The yellow arrow represents the Burgers circuit. The white circle indicates the position of the screw dislocation. Scale bar: 2 nm. The color scale represents normalized strain values from compressive (negative) to tensile (positive). Source data are provided as a Source Data file.

## Dislocation multiplication in curved space

The coupling of 3D printing with screw dislocation not only effectively mitigates the heterointerface issues in conventional electrocatalysts, but also provides a pathway to achieve tunable 3D lattice strain engineering. Traditionally, ideal screw dislocation growing at a flat substrate normally exhibits an aligned spiral shape, with all edges stacking in parallel[22,23,31]. However, we observed a unique twisting morphology in certain regions (Fig. 1e–g), which is rare for macroscale

crystals[19]. The elastic instability in curved space may be a primary reason for this crystal geometric frustration[32]. The curved surface of shell-lattice metamaterials introduces a non-Euclidean geometry, compelling growth layers to conform to a curved surface, which results in inevitable stretching or compression of lattice bonds. Consequently, the curved substrate introduces additional geometrical incompatibility elastic energy, highly dependent on underlying curvature. Absolute Gaussian curvature $K$ was employed to delineate the geometric properties of a surface at a given point, derived from the absolute product of two principal curvatures, $k_1$ and $k_2$, also referred to as maximum and minimum curvature, respectively[33]. In the case of a disk-shaped layered crystal of diameter $a$ on a spherical substrate of radius $R$, the elastic strain energy can be expressed as:

$$E = \frac{b^2\mu}{4\pi} ln \frac{a}{2} + \frac{Y\pi}{384} \frac{a^6}{R^4} \qquad (1)$$

where $Y$ is the Young's modulus of the crystal, $\mu$ is the shear modulus, **b** is the Burgers vector of the screw dislocation. On conventional flat substrates (where $R$ tends to infinity or curvature approaches zero), the elastic strain energy is solely a result of the screw dislocation. The strain energy for micrometer-sized bulk crystals is typically negligible due to their large crystal diameter $a$ (Fig. 3a), allowing them to retain global translational symmetry across each layer. However, on curved substrates, the introduction of curvature and frustration generates additional elastic energy, with its magnitude increasing remarkably as the domain size exceeds a critical threshold $a^*$ (Fig. 3a). Once accumulated elastic strain energy surpasses a critical threshold, it can induce geometric frustration as the system attempts to accommodate the surface stress[34]. Under this condition, isotropic growth becomes energetically unfavorable, leading to the formation of twisted nanosteps that align with changes in the substrate curvature (Supplementary Fig. 13).

The high elastic strain energy at twisted nanosteps can be released by destabilizing the symmetry of the crystal through dislocation multiplication. Consequently, we observed a gradual increase in dislocation density with the increased curvature of the substrate (Fig. 3b). Regarding the type of dislocations, the statistical analysis results (Supplementary Fig. 14a, b) reveal that the percentage of edge dislocations gradually increases from 22% to 38% as the substrate curvature increases. Nevertheless, screw dislocations remain the dominant type even in areas with large curvature. To quantify the influence of dislocation multiplication on the matrix lattice, we measured the matrix lattice spacing of the step edge from HRTEM image (Fig. 3b). The (111) lattice spacing was measured as 2.07 Å, 2.13 Å, and 2.20 Å with increasing of localized curvature, corresponding to lattice tensions of approximately 0%, 2.9%, and 6.3%, respectively (Supplementary Fig. 14c). Using calculated absolute Gaussian curvature mapping, we qualitatively determined the lattice strain distribution on the surface of a typical unit cell in shell-lattice metamaterials (Fig. 3c), showing a positive correlation with Gaussian curvature. For comparative analysis, we fabricated two additional reference metamaterials—simple cubic lattice and single-order gyroid—alongside the dual-scale gyroid. To enhance comparability, they maintained consistent relative densities of ~7% and specific surface areas of ~2.5 mm⁻¹, while achieving significantly different mean absolute Gaussian curvatures (measured at 0, 192, and 4650 mm⁻², denoted as S0, S192, and S4650, respectively, as shown in Fig. 3d). XRD results in Fig. 3e and Supplementary Fig. 15 revealed that increasing mean Gaussian curvatures shifts the typical (200) diffraction peaks to lower angles, indicating an increase in crystal lattice constant. Williamson–Hall calculation (Supplementary Fig. 16) revealed that FeCoNi-S4650 ($2.23 \times 10^{14}$ m⁻²) displayed remarkably higher dislocation density compared to FeCoNi-S0 ($0.84 \times 10^{14}$ m⁻²).

## Electrocatalysis performance and mechanism

We evaluated the performance of dislocation-rich metamaterials in nitrate-to-ammonia reduction reaction ($NO_3RR$). According to the linear sweep voltammetry (LSV) results, the reduction of $NO_3^-$ ions was primarily attributed to the enhanced current density, while the competing hydrogen evolution reaction (HER), which delivers a nearly zero current density across a potential range of −0.1-−0.6 V vs. RHE, indicating a high selectivity toward $NO_3RR$ within this broad potential range (Supplementary Fig. 17)[35]. LSV results (Supplementary Fig. 18) revealed that incorporating the nanosteps structure and dense dislocation network into metamaterials can remarkably enhance current density over a wide range of potentials. The electrochemically active surface area (ECSA) was measured to accurately derive the intrinsic activity of these catalysts (Supplementary Fig. 19)[36]. The ECSA-normalized current density for $NO_3RR$ ($j_{ECSA}$) (Fig. 4a) revealed that FeCoNi-S4650 (dual-scale gyroid with screw dislocation growth) with the highest dislocation density, exhibited the highest current density. Similarly, Faraday efficiency (FE) and $NH_3$ yield rate ($Y_{NH3}$) followed the same trend with increasing dislocation density (Supplementary Fig. 20). Remarkably, FeCoNi-S4650 achieved the highest $FE_{NH3}$ of 95.4% and the highest $Y_{NH3}$ of 20.58 mg h⁻¹ cm⁻², nearly 5 times higher than FeCoNi-LBL4650 (dual-scale gyroid with the LBL growth) (Fig. 4b). Moreover, the lowest $FE_{NO2^-}$ suggests that introducing a high-density dislocation network improves both the catalytic activity and selectivity for $NH_3$. Based on the LSV results, we further studied the effect of applied voltage on $NH_3$ selectivity. Specifically, FeCoNi-S4650 achieved high $FE_{NH3}$ (>90%) at a lower applied potential than FeCoNi-LBL4650, and this efficiency was sustained across a broad potential window from −0.5 to −0.7 V (Supplementary Fig. 21a). We also calculated energy efficiency (EE) for FeCoNi-S4650 and FeCoNi-LBL4650 at various applied potentials. The analysis confirms that FeCoNi-S4650 achieved higher EE than FeCoNi-LBL4650 across a broad potential window of −0.1 to −1.1 V. Notably, FeCoNi-S4650 exhibited a peak $NH_3$ cathodic half-cell EE of 30.3% at an applied potential of −0.5 V (Supplementary Fig. 21b). pH monitoring (Supplementary Fig. 21c, d) for FeCoNi-S4650 and FeCoNi-LBL4650 catalyst showed catholyte pH rises within 1 hour at our applied potential, confirming that proton transport was insufficient to prevent excessive alkalization. These results indicate that the metamaterial catalysts with dense dislocation networks promote $NO_3^-$-to-$NH_3$ conversion at high efficiency and lower energy consumption, positioning them as promising candidates for the economically feasible synthesis of $NH_3$. Amperometric (i-t) analyses revealed that the FeCoNi medium-entropy alloy (MEA) achieved markedly higher $NH_3$ Faradaic efficiency and production rates compared to monometallic and binary systems (Supplementary Fig. 22). Moreover, we conducted density functional theory (DFT) calculations to further study the synergistic effects induced by multi-elemental mixing in MEA optimize the electrochemical nitrate reduction process (Supplementary Note 3 and Supplementary Figs. 23,24).

To confirm the source of the N-containing products, isotope labeling ¹H NMR spectra were performed. With $^{14}NO_3^-$/$^{15}NO_3^-$ in a ratio of 1 to 1 as the feeding source, the ratio of generated $^{14}NH_4^+$/$^{15}NH_4^+$ remains almost unchanged. This confirms that $NH_3$ production originates from the $NO_3RR$ rather than external environmental pollutants (Supplementary Fig. 25). Meanwhile, the ¹H NMR spectra showed trimodal peaks of $^{14}NH_4^+$ when $^{14}NO_3^-$ was used as the feed (Supplementary Fig. 26a). $NH_3$ FE calculated from both NMR and UV-Vis methods were in close agreement, validating the accuracy of these two quantitative techniques (Supplementary Figs. 26b, 27-28).

Apart from the outstanding catalytic activity, electrocatalytic stability is essential for potential industrial application. Therefore, long-term stability tests were performed over 21 consecutive electrolysis cycles, each lasting 24 hours (Fig. 4c). Even after more than 500 hours of operation, the FeCoNi-S4650 catalyst maintained an $FE_{NH3}$ of 80%, demonstrating its high electrochemical stability. The

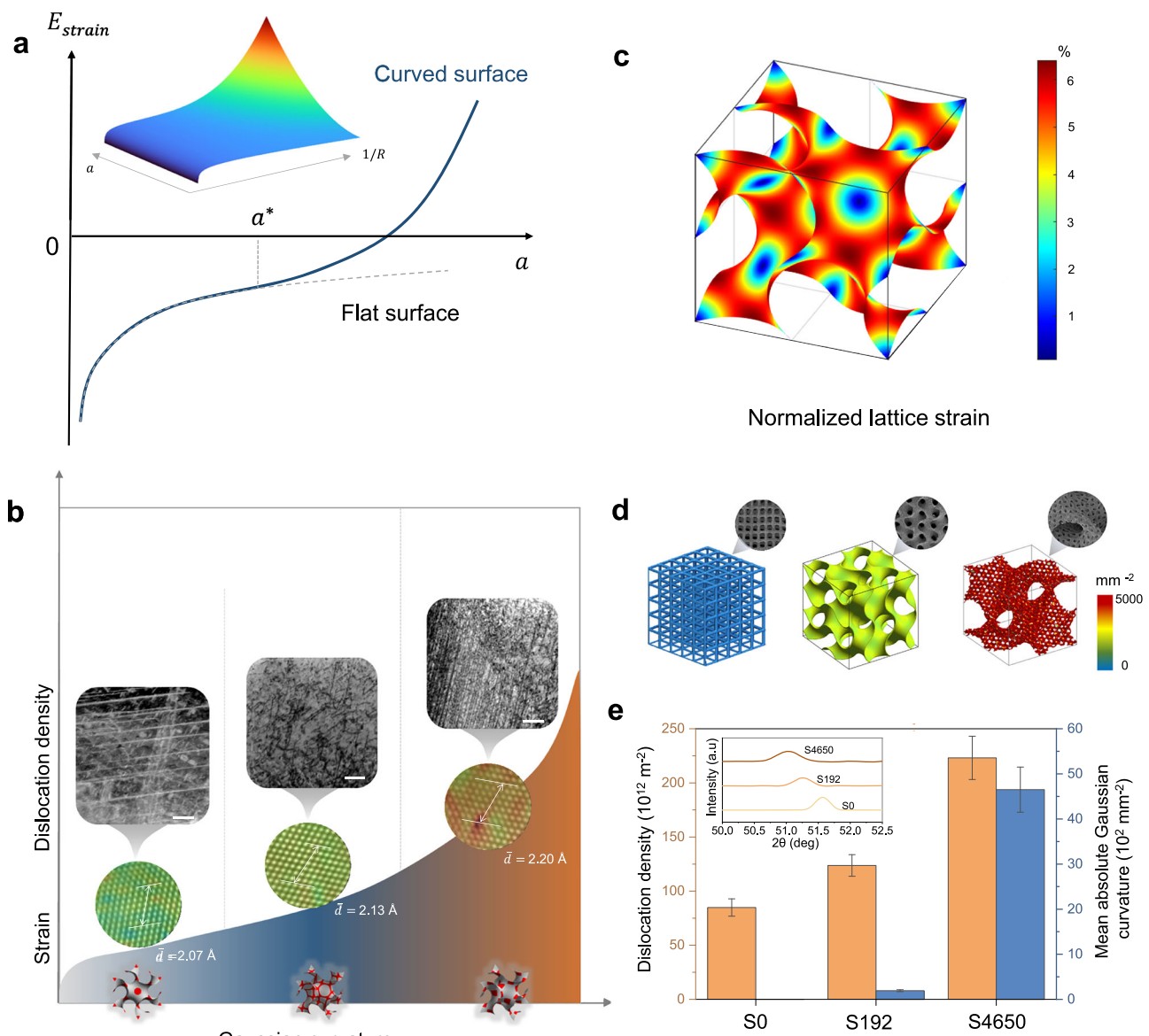

**Fig. 3 | Dislocation multiplication in curved space. a** Schematic elastic strain energy for growth of a crystal with diameter $a$ on a flat surface (gray line) and a curved surface (blue line). Insert displays the relationship between elastic strain energy as a function of crystal diameter $a$ and Gaussian curvature $1/R$. **b** Influence of substrate Gaussian curvature on the dislocation density and lattice strain at surface nanosteps. The scale bar is 100 nm for inserted scanning transmission electron microscopy (STEM) images. $\bar{d}$ indicates the average lattice spacing for inserted HRTEM images. **c** Normalized lattice strain distribution of a gyroid unit cell. **d** Quantification and distribution of mean absolute Gaussian curvatures of the designed scaffolds. **e** The comparison of surface dislocation density and mean Gaussian curvatures for different scaffolds. The inserted intensity profiles of XRD display a gradual left-direction shift, demonstrating increased lattice strain. The error bars indicate the standard deviations calculated from three replicate measurements. Source data are provided as a Source Data file.

post-mortem SEM and XRD characterizations showed negligible changes in the morphology and composition (Supplementary Fig. 29), further demonstrating the long-term stability. While slight fluctuations in current density observed during the cycling tests can be attributed to the "disentanglement" process of dislocations (Supplementary Fig. 30), leading to the typical strain relaxation phenomenon[37,38]. Such long-lasting electrocatalytic performance outperforms most reported Cu-based, Fe-based, and Co-based $NO_3RR$ catalysts (Supplementary Fig. 31 and Supplementary Table 3).

The $NO_3^-$-to-$NH_3$ reaction is a complicated 8-electron transfer process, resulting in a variety of byproducts and multiple possible pathways. Online differential electrochemical mass spectrometry (DEMS) and in situ Raman spectroscopy were carried out to detect the molecular intermediates and products over the FeCoNi-S4650 (Fig. 4d, e and Supplementary Fig. 32). Due to the page limitation,

detailed discussions of DEMS and in situ Raman spectroscopy results are shown in Supplementary Note 4. Based on those combined analysis, the overall roadmap of $NO_3RR$ on FeCoNi-S4650 was proposed. $NO_3RR$ pathway involves the adsorption of *$NO_3$, followed by a series of deoxidation (*$NO_3$ → *$NO_2$ → *$NO$) and hydrogenation reactions (*$NOH$ → *$NH_2O$ → *$NH_2OH$ → *$NH_2$ → *$NH_3$), and ultimately desorption of *$NH_3$ from the catalyst surface.

To elucidate the mechanism behind the enhanced $NO_3RR$ activity, we performed theoretical calculations on the metamaterial catalysts, characterized by the surface step defects and inhomogeneous lattice strain induced by dense dislocation network (Fig. 5a). Thermodynamically, strong affinity for $NO_3^-$ is a critical prerequisite for efficient $NO_3^-$-to-$NH_3$ conversion[39]. Figure 5b and Supplementary Figs. 33–36 summarize the calculated values of Gibbs free energies ($\Delta G_{*NO3}$). FeCoNi catalyst presents more active sites than the single

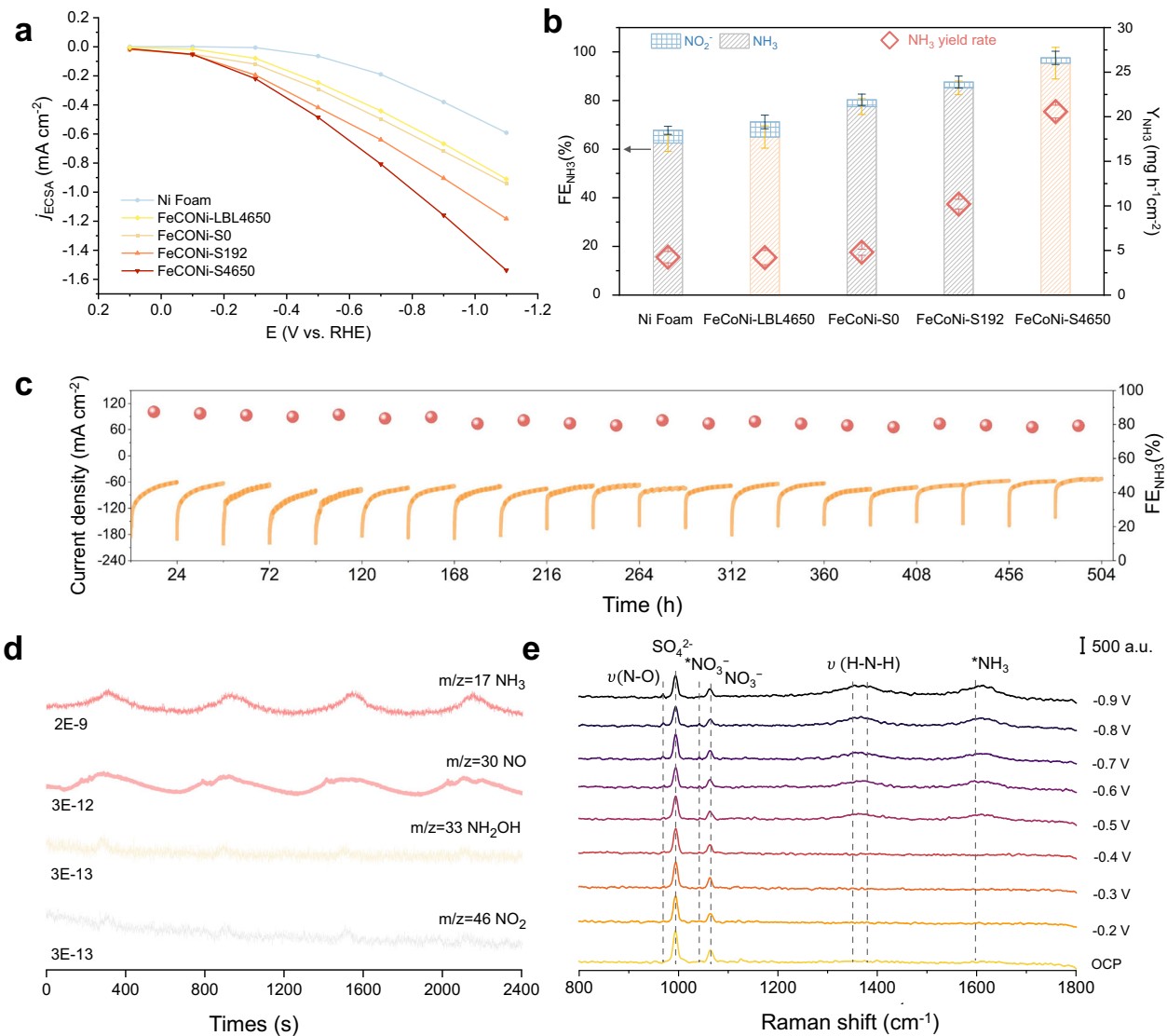

**Fig. 4 | NO₃RR performance of dislocation-rich metamaterial catalysts. a** ECSA normalized partial current densities in LSV curves of Ni foam, FeCONi-LBL4650, FeCoNi-S0, FeCoNi-S192, and FeCoNi-S4650 in an electrolyte of 0.5 M Na₂SO₄ and 0.1 M NaNO₃ with resistance compensation. Resistances (ohm): 2.9 ± 0.02 (Ni foam), 2.7 ± 0.04 (FeCoNi-LBL4650), 2.5 ± 0.03 (FeCoNi-S0), 2.3 ± 0.05 (FeCoNi-S192), and 2.6 ± 0.03 (FeCoNi-S4650). **b** Faraday efficiency (FE_NH3) and NH₃ yield rate (Y_NH3) of Ni foam, FeCoNi-LBL4650, FeCoNi-S0, FeCoNi-S192, and FeCoNi-S4650. Error bars represent the standard error of the mean derived from three independent replicate measurements. **c** Time-dependent current density curves and corresponding FE_NH3 of the FeCoNi-S4650 cathode in consecutive 21 recycling tests (each test lasting 24 h). **d** in situ DEMS measurements of FeCoNi-S4650 for nitrate electroreduction. **e** in situ Raman spectra of NO₃RR over FeCoNi-S4650 at different applied potentials. Source data are provided as a Source Data file.

metal slab. The nine adsorption sites of the FeCoNi-LBL slab exhibit a wide range of $\Delta G_{*NO3}$ values, spanning from 0.68 to −0.72 eV. Active sites on stepped FeCoNi contribute to lower $\Delta G_{*NO3}$ values from 0.54 to −1.12 eV, indicating the variation in atomic coordination induced by screw dislocation can drastically enhance the adsorption of NO₃⁻. Subsequently, the lattice tensile strain from the high-density dislocation network was introduced in a stepped FeCoNi slab (Fig. 5b). The stepped FeCoNi under 3% tensile strain exhibits optimized $\Delta G_{*NO3}$ values ranging from 0.36 to −1.21 eV. Notably, at 6% tensile strain, NO₃⁻ adsorption is further enhanced across various active sites, with Gibbs free energy changes ranging from −0.01 to −1.26 eV. Specifically, the bridge site of Fe-Fe contributes to the most favorable NO₃⁻ adsorption, suggesting that strong adsorption of NO₃ lowers the energy barrier of the initial discharge step (*+NO₃ → *NO₃ + e⁻) (Fig. 5b).

Furthermore, the activation of *NO₃ on the Fe-Fe site was analyzed by differential charge distribution and the partial density of states. From the charge density difference (Fig. 5c), nanosteps defect arising from screw dislocation model significantly improves the charge transfer from Fe-Fe atoms to NO₃⁻ (0.69 e⁻ and 0.72 e⁻ for FeCoNi-LBL and stepped FeCoNi, respectively). With the introduction of lattice strain, stepped FeCoNi with 6% strain shows the highest charge transfer value (0.83 e⁻), offering strong evidence of NO₃⁻ adsorption. This enhanced interaction can be understood through the d-band center model, which suggests that the stepped configuration and lattice strain in FeCoNi-LBL slab induce an upward shift of 3d-band center (Fig. 5d). Subsequently, the deoxidation and hydrogenation steps proceed, with Gibbs free energies of each intermediate analyzed at the strongest NO₃⁻ adsorption site (the

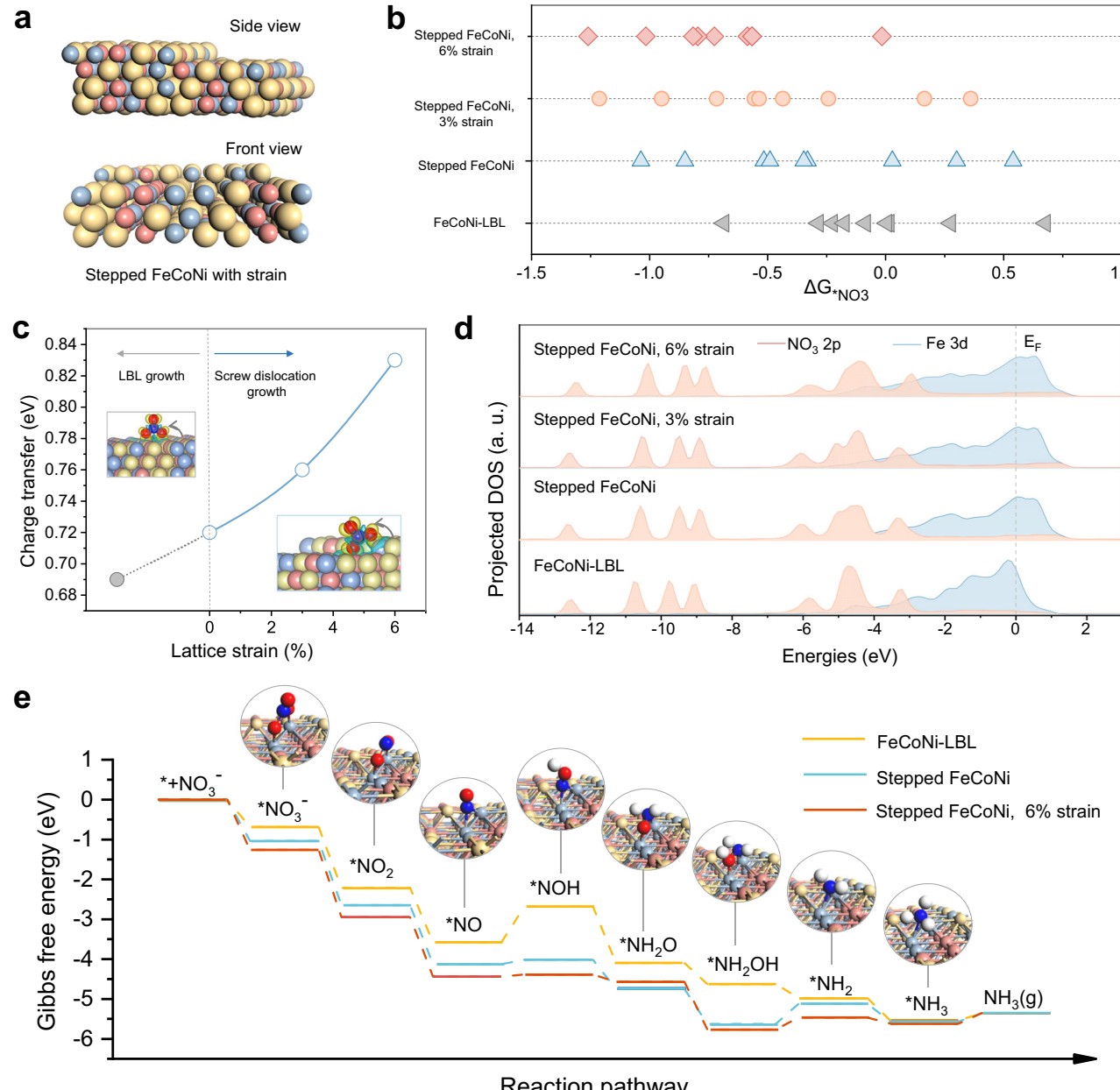

**Fig. 5 | Theoretical analysis of the catalytic mechanism. a** Side view and front view of stepped FeCoNi calculation model, where the yellow balls represent Fe atoms, the blue balls represent Co atoms, and the red balls represent Ni atoms. **b** The *NO₃ adsorption Gibbs free energy ($\Delta G_{*NO3}$) of FeCoNi-LBL, stepped FeCoNi, and stepped FeCoNi with 3% and 6% strain models, where the synergic effects of nanosteps configuration and lattice strain on $\Delta G_{*NO3}$ are investigated. **c** The charge density difference of LBL growth and screw dislocation growth as a function of lattice strains. **d** The projected electronic density of states for d orbital of FeCoNi-LBL, stepped FeCoNi, and stepped FeCoNi with 3% and 6% strain models. **e** Free-energy diagram for NO₃RR on FeCoNi-LBL, stepped FeCoNi, and stepped FeCoNi with 6% strain. Source data are provided as a Source Data file.

bridge site of Fe-Fe) (Fig. 5e and Supplementary Fig. 37, 38). FeCoNi-LBL and stepped FeCoNi exhibit various uphill free energy increments in the electron-proton transfer. The *NO hydrogenation step (*NO + H₂O + e⁻ → *NOH + OH⁻) was the potential-determining step (PDS), involving a ΔG of 0.89 eV over FeCoNi-LBL. In contrast, the PDS on the stepped FeCoNi slab is the *NH₂OH to *NH₂ step, with a significantly lower thermodynamic barrier of 0.51 eV. This PDS change is due to the exposed step sites created by screw dislocation, which reduce the free energy of *NO to *NOH from 0.89 eV to 0.11 eV. Moreover, lattice strain in stepped FeCoNi slab retains the same PDS (*NH₂OH to *NH₂) while further advancing the progress of NO₃⁻ reduction with a lower energy barrier of 0.3 eV. The lower

thermodynamic barrier of NO₃RR could markedly assist the following deoxidation/hydrogenation steps in NO₃RR. Therefore, introducing screw dislocation can enhance the catalytic activity for NO₃RR, potentially promoting NH₃ synthesis. Although screw dislocations remain the predominant type across substrates with varying curvatures, the role of edge dislocations should also be studied in modulating catalytic activity. According to the density functional theory calculation of edge dislocations (Supplementary Fig. 39), the potential-determining step on the FeCoNi slab with edge dislocations is the *NH₂OH to *NH₂ step, with a significantly lower thermodynamic barrier of 0.65 eV. These exposed sites created by edge dislocations reduce the energy barrier of NO₃RR as compared to FeCoNi-LBL.

## Practical applications of metamaterial catalysts

Mechanical stability is a key indicator in evaluating the industrial applicability of electrocatalysts, however, existing research efforts appear disproportionately focused on performance optimization, with insufficient attention given to this essential aspect. Conventionally, the mechanical stability of electrocatalysts mainly involves the interface adhesion strength between support and nanomaterials and the intrinsic strength of porous catalyst support. On the one hand, the impact and drag forces induced by electrolyte convection and turbulence can cause catalyst detachment, which is a common problem, especially for conventional powdery catalysts[40]. On the other hand, during the industry-level operating environment, the porous catalyst support is susceptible to diverse external compressive forces exerted by the electrode clips, water pressure, and adjacent components, leading to a diminution of its operational lifespan[16]. To address this, it is necessary to further investigate the mechanical properties of our optimized FeCoNi dual-scale gyroid, especially compared to commercial Ni foam (a commonly used support material) at an equivalent relative density of ~7.0 %.

Integrating the advantage of architecture and material (Supplementary Note 5 and Supplementary Fig. 40–42), our FeCoNi dual-scale gyroid not only demonstrates significantly higher strength compared to Ni foam, with a compression strength 3 times higher (Fig. 6a), but also exhibit a more homogeneous deformation. In situ SEM images display a near layer-by-layer collapse with little to no thin-wall fracture over 50% strain (Fig. 6b–c and Supplementary Movie 1). In contrast, commercial Ni foam displayed low mechanical strength and typical strain localization. Traditionally, alumina and carbon foam have been extensively employed as electrocatalyst support materials for nanoparticles in the automotive industry[41,42]. However, these foams consist of interconnected beams or struts which not only induce stress concentration at its joints or nodes, but also accommodate compressive strain via bending of its struts, leading to a low specific strength (Fig. 6d). Furthermore, the formation of shear bands in the Ni foam localizes deformation to a particular region rather than being distributed throughout the entire sample, thereby resulting in moderate energy absorption capability (Fig. 6e). The "nodeless" nature of TPMS architectures allows it to distribute stress more evenly, minimizing localized deformation and resulting in both superior specific strength and energy absorption. In addition to the contributions from architectural design, the optimization of the multicomponent composition also played a significant role in enhancing both strength and ductility. With the same dual-scale gyroid design, FeCoNi sample still exhibited 2 times higher strength than pure Ni and displayed ductile fracture surface morphologies with numerous dimples (Supplementary Fig. 41). Overall, FeCoNi dual-scale gyroid achieved an outstanding balance of mechanical properties by integrating the advantage of shell-based architecture and multicomponent composition design, outperforming most of the previous reported foam structures or lattice structures (Fig. 6d–e)[43-51]. Post-mortem SEM images (Supplementary Fig. 42) showed that microcracks remained confined to the "weak" grain boundary areas, without propagating into the interior nanostructures. This indicates that the integrated metamaterial catalysts provide exceptional bonding strength between the support and surface nanostructures, effectively preventing peeling of nanomaterials under external forces.

To demonstrate the practical applicability of metamaterial catalysts, we designed an integrated three-chamber reactor assembled with the flow-through $NO_3^-$ reduction electrolyzer and a membrane-based ammonia absorption unit to continuously produce and capture ammonia products, as illustrated in Fig. 6f and experimentally demonstrated in Supplementary Fig. 43. Using this device under industrial-relevant current densities, the integrated metamaterial catalyst achieved over 200 h of long-term stability at ~350 mA cm⁻² with over 90% $FE_{NH3}$ maintained (Fig. 6 g), confirming its applicability at an industrial level. During the electrocatalysis process, in situ generated $NH_3$ spontaneously permeates across the gas exchange membrane to the recovery chamber, capturing it by the flowing HCl solution with a collection efficiency of over 91.3%. After conducting rotary evaporation, the resulting $NH_4Cl$ effluent in the acid absorption chamber is ultimately converted into high-purity $NH_4Cl$ powder (Supplementary Fig. 44). The recycled $NH_4Cl$ was confirmed by the XRD pattern (Fig. 6h), which can be used as fertilizer and other downstream applications. Moreover, we calculated the energy consumption for the combined electrochemical $NO_3^-$-to-$NH_3$ conversion and nitrate reduction process, and the result was 25.6 kWh kg⁻¹ N. These findings demonstrated a profitable and environmentally friendly conversion route that directly recovers upgraded ammonia fertilizers from nitrate-containing wastewater using the assembly device, leveraging the shell-lattice metamaterials. Overall, our 3D-printed metamaterial catalysts, driven by screw dislocations, integrate multiple advantages, including design flexibility, structural strength, catalytic activity, and outstanding bonding strength, outperforming other 3D printing strategies[16,17,52,53] and conventional randomly porous catalysts[54] (Fig. 6i).

## Discussion

In summary, we introduce a scalable, integrated 3D printing strategy that incorporates screw dislocations under low supersaturation conditions to achieve hierarchical metamaterial catalyst, featuring interface-free surface nanostructures and high dislocation storage. The simultaneous synthesis of curved substrates and surface nanostructures ensures strong interface bonding and maximizes elastic strain energy accumulation during the growth process, which is released by introducing dislocation defects to destabilize the crystal symmetry. Therefore, we achieved tunable 3D lattice strain engineering via metamaterial curvature control. The step defects and inhomogeneous lattice strain can effectively enhance the electrocatalytic activity for NO₃RR by modulating $NO_3^-$ adsorption and lowering the reaction energy barrier. Finally, by coupling nitrate reduction with acid absorption, we designed an integrated three-chamber reactor to continuously produce and capture ammonia products with over 200 h of long-term stability at ~350 mA cm⁻² while maintaining more than 90% $FE_{NH3}$, highlighting its potential for industrial applications.

## Methods

### Materials and chemicals

Dimethyl sulfoxide (DMSO, 99.7%), Poly (ethylene glycol) diacrylate (PEG, average $M_n = 700$), and 1-(phenyldiazenyl) naphthalen-2-ol (Sudan I, Dye content ≥95 %) were purchased from Sigma-Aldrich. Phenylbis (2,4,6-trimethylbenzoyl) phosphine oxide (BAPO, 99.7%) was purchased from Aladdin. Metal salt used in this work, including iron nitrate nonahydrate (99.99%), cobalt nitrate hexahydrate (99.99%), and nickel nitrate hexahydrate (99.99%), were purchased from Sigma-Aldrich. Sodium sulfate (99.99%), sodium nitrate (99.99%), hydrochloric acid (37%), deuterium oxide ($D_2O$, 99.9%), and potassium nitrate-¹⁵N (98 atom% ¹⁵N) were purchased from Sigma-Aldrich. Pure hydrogen gas (Purity: 99.999%) and pure argon gas (Purity: 99.995%) were purchased from Linde HKO Ltd (Hong Kong). Commercial Ni foams were purchased from Kunshan Xingzhenghong Electronic Materials Co., Ltd in China.

### Preparation of integrated metamaterials catalyst

To prepare the 3D printing resin, 70 ml of DMSO was first mixed with 10 ml PEG diacrylate. 600 mg BAPO photoinitiator (Aladdin, >97%) and 25 mg Sudan I were uniformly stirred into the DMSO/PEG mixture solution. Then, 3D architected organogel scaffolds were fabricated layer by layer using a DLP 3D printer (Phrozen Sonic mini 8k) with a 405 nm light source. The slicing distance was 50 μm, with a curing time of 18 s for each layer. Subsequently, architectures were soaked in

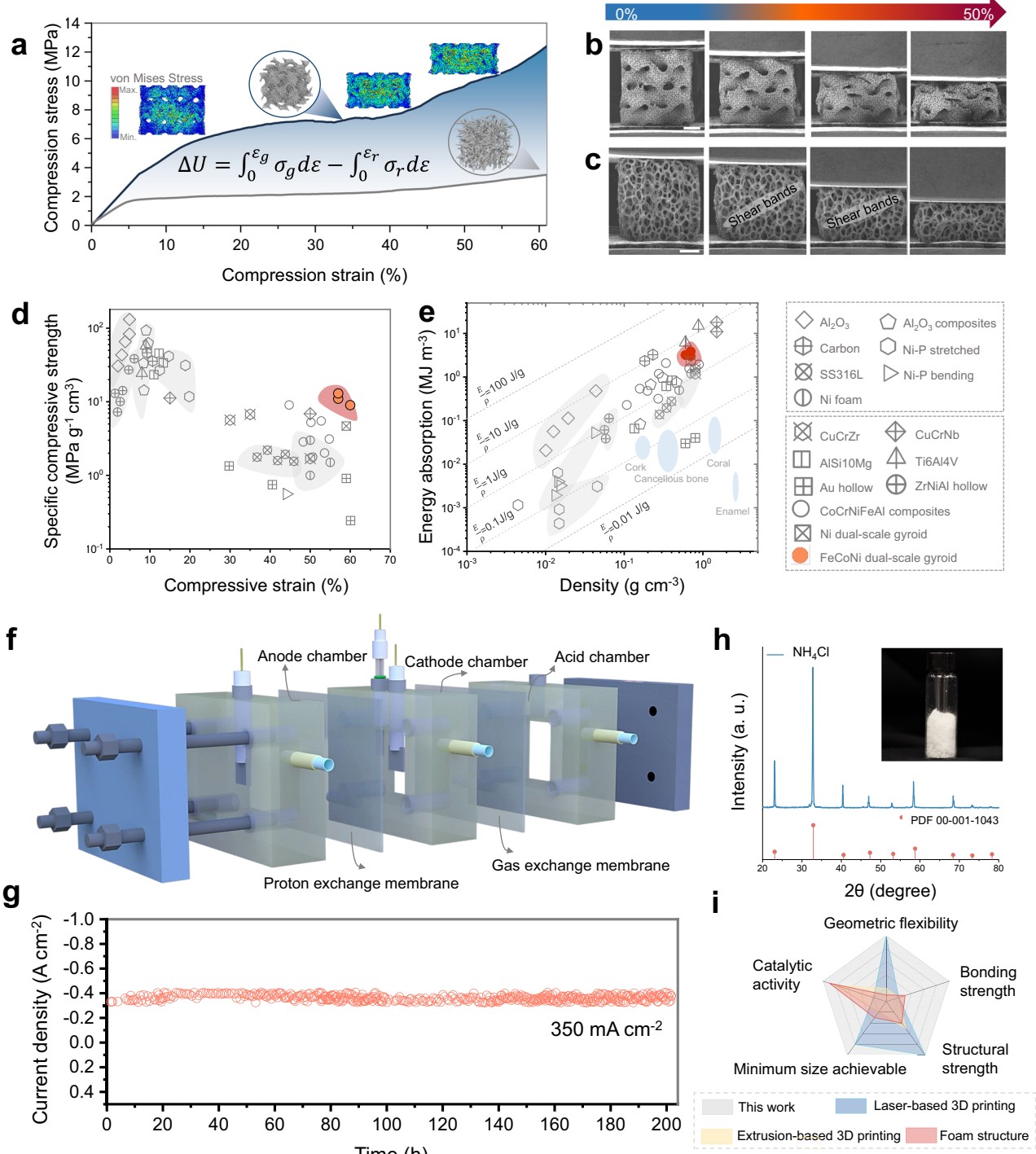

**Fig. 6 | Mechanical performance and ammonia product collection.**
**a** Compression stress-strain curve of dual-scale gyroid and commercial Ni foam with a consistent relative density of ~7%. The inserted finite element analysis result shows the stress distribution during deformation. The color scale indicates the von Mises stress range. in situ SEM uniaxial compression deformation experiment revealing the homogeneous deformation behavior of **b** FeCoNi dual-scale gyroid metamaterials and **c** Ni foam. Scale bar: 1 mm. **d** Ashby map of specific compression strength versus density and **e** energy absorption per unit volume versus density of dual-scale gyroid metamaterials compared to reported metamaterials[43–51]. **f** Schematical illustration of a three-chamber reactor assembled with the flow-through $NO_3^-$ reduction electrolyzer and a membrane-based ammonia absorption unit. **g** Long-term stability test of metamaterials catalyst. **h** Generated $NH_4Cl$ powder and corresponding XRD result. **i** Typical characteristics of different 3D-printed catalytic electrodes and conventional foam electrodes. Source data are provided as a Source Data file.

deionized water for 6 h at room temperature to convert to hydrogel scaffolds. 2 M and 10 M metal salt solution of nickel nitrate, iron nitrate, and cobalt nitrate for FeCoNi alloy were prepared with deionized water. Hydrogel architectures were immersed in a corresponding metal salt solution for 12 h then taken out and put in a vacuum oven for preliminary drying at 50 °C for 48 h. Then, shell-lattice metamaterials were calcined in an air tube furnace with a maximum temperature of 800 °C to obtain the metal oxide structures.

Finally, the FeCoNi shell-lattice metamaterials were obtained by reducing the corresponding metallic oxide in an $H_2$: Ar = 30:70 and 100 sccm flow at 1000 °C. Different reduction durations (6 h, 8 h, and 10 h) were chosen to evaluate the influence on the growth mode. To conduct the comparison experiment, we further fabricated a series of dual-scale shell metamaterial catalysts with different composition components using the same process and geometry parameters, including Fe, Co, Ni, FeCo, FeNi, and CoNi.

## Microstructure characterization

The geometry and microstructure of the integrated metamaterial catalysts were observed using field emission SEM (FESEM, FEI Quanta 450) equipped operated at 20-kV acceleration voltage. The chemical composition and dislocation density of the catalyst were analyzed using SEM with energy-dispersive X-ray spectroscopy (EDS) and electron backscatter diffraction (EBSD) detectors. A transmission electron microscope (TEM, JEOL JEM 2100) equipped with SAED was employed to observe the microstructure operating at 200 KV, where the TEM sample was prepared via focused ion beam (FIB, FEI Scios DualBeam). STEM high-angle annular dark field (STEM-HAADF) analysis was performed in a Thermo Scientific Talos F200X TEM with an accelerating voltage of 200 KV. X-ray diffraction (XRD, Rigaku SmartLab) was employed to investigate the crystal structure and dislocation density. For EBSD characterization, porous structures were electrochemically polished using a solution of $HNO_3$:$C_2H_6O$ in a ratio of 1:4. AFM was employed to measure the height and width of nanosteps using a Bruker Icon AFM in tapping mode using OTESPA-R3 tips from Bruker AFM Probes.

The APT characterizations were performed in a local electrode atom probe (CAMEACA LEAP 5000 XR). The specimens were analyzed at 70 K in voltage mode, at a pulse repetition rate of 200 kHz, a pulse fraction of 20%, and an evaporation detection rate of 0.2% atom per pulse. The data analysis workstations, AP Suite 6.3 were used to create 3D reconstructions and data analysis.

## Electrochemical measurements

All electrochemical tests were conducted in a typical H–type cell separated by a Nafion 117 membrane, where the FeCoNi (10 mm × 5 mm × 2 mm), Ag/AgCl, and platinum foil were designated as the working, reference, and counter electrodes, respectively. All the potentials were converted to the RHE reference scale by $E_{RHE} = E_{Ag/Agcl} + 0.059 \times pH + E^0_{Ag/Agcl}$ ($E^0_{Ag/Agcl} = 0.197$). Nafion 117 membrane (thickness 183 μm, 3 cm × 3 cm) was pretreated by sequential boiling in deionized water, 3% $H_2O_2$, and deionized water (1 h each) to ensure proton conductivity. The Ag/AgCl (3 M KCl) electrode was calibrated before measurements against a reversible hydrogen electrode (RHE) in 0.1 M $HClO_4$ at 25 °C, showing a stable potential of 0.210 V vs. RHE. Unless otherwise stated, all electrode areas are the geometric areas of the electrodes. Electrochemical measurements employed ohmic drop compensation, with solution resistances determined via the iR Comp module (CHI 660E workstation) through at least three independent measurements prior to testing. Before potentiostatic measurements, linear sweep voltammetry (LSV) was performed at a rate of 5 mV s$^{-1}$ in a potential range between 1 and −1.1 V vs. RHE. Nitrate electroreduction experiments entailed conducting potential-controlled measurements for 1 hour at a stirring rate of 600 rpm for each potential. Three replicate electrochemical experiments were performed on each sample to establish an error bar. A 0.5 M $Na_2SO_4$ solution (40 mL) served as both the catholyte and the anolyte. Additionally, 0.1 M of nitrate-N was introduced into the cathode compartment for the electrochemical nitrate reduction test. Prior to each experiment, fresh electrolyte was prepared and purged with argon for 10 minutes to remove dissolved oxygen and nitrogen. Amperometric (i-t) analyses and the long-term stability tests were performed at a fixed potential of −0.7 V vs. RHE. The electrochemically active surface area (ECSA) was calculated using the double-layer capacitance (Cdl), determined from the cyclic voltammetry curves within a specific potential range where faradaic current was absent. The working electrode underwent scanning in a 1.0 M KOH solution at sweep rates ranging from 20, 40, 60, 80, and 100 mV s$^{-1}$, covering a potential range from 0.7 to 0.85 V. The current density plotted against the scan rate displayed a linear correlation at a specific potential, with the slope representing the double-layer capacitance Cdl (mF cm$^{-2}$).

## Computational methods

This investigation leverages first-principles DFT computations implemented within the Vienna ab initio Simulation Package (VASP) platform. Valence electron interactions were treated using the projector augmented-wave (PAW) pseudopotential approach, while electron exchange-correlation effects were described through the Perdew-Burke-Ernzerhof (PBE) functional within the generalized gradient approximation (GGA) framework[55,56]. Localization of iron 3d electrons was modeled using the DFT + U approach based on the Hubbard Hamiltonian. Different models were constructed based on the FeCoNi structure. For the model of the step structure of FeCoNi caused by screw dislocations, 4-layered (200) slabs of FeCoNi were modeled. A 15 Å vacuum spacing was implemented to mitigate inter-slab interactions, complemented by a Monkhorst-Pack 3 × 3 × 1 k-point sampling scheme for reciprocal space integration. Electronic structure calculations employed a 500 eV plane-wave cutoff energy, with electronic self-consistency to 10$^{-5}$ eV accuracy and atomic force convergence below 0.02 eV Å$^{-1}$ within the Hellmann-Feynman framework. VESTA and VASPKIT were used to obtain the DOS diagrams[57].

The nitrate reduction pathway proceeds through these sequential steps (2–10):

$$* + NO_3^-(l) \rightarrow *NO_3 + e^- \tag{2}$$

$$*NO_3 + H_2O + 2e^- \rightarrow *NO_2 + 2OH^- \tag{3}$$

$$*NO_2 + H_2O + 2e^- \rightarrow *NO + 2OH^- \tag{4}$$

$$*NO + H_2O + e^- \rightarrow *NOH + OH^- \tag{5}$$

$$*NOH + H_2O + e^- \rightarrow *NH_2O + OH^- \tag{6}$$

$$*NH_2O + H_2O + e^- \rightarrow *NH_2OH + OH^- \tag{7}$$

$$*NH_2OH + e^- \rightarrow *NH_2 + OH^- \tag{8}$$

$$*NH_2 + H_2O + e^- \rightarrow *NH_3 + OH^- \tag{9}$$

$$*NH_3 \rightarrow * + NH_3 \tag{10}$$

Where the * represents the active site.

For each step, the reaction free energy was determined using the following Eq. (11):

$$\Delta G = \Delta E + \Delta E_{ZPE} - T\Delta S \tag{11}$$

Here, ΔE represents the total energy difference between reactant and product states. $\Delta E_{ZPE}$ and ΔS denote the corresponding changes in zero-point energy and entropy, derived from vibrational frequency calculations, while T is the temperature (298.15 K).

To circumvent the direct calculation of the charged $NO_3^-$ energy, neutral $HNO_3$ in the gas phase was employed as the reference, as shown in the equation below (12, 13)[58,59]:

$$NO_3^-(l) + H^+ \rightarrow HNO_3(l) \tag{12}$$

$$HNO_3(l) \rightarrow HNO_3(g) \tag{13}$$

Hence, the Eq. (2) can be rewritten as below (14):

$$* + HNO_3(g) \rightarrow *NO_3 + H^+ + e^- \tag{14}$$

The Gibbs free energy of adsorbed nitrate ($\Delta G(*NO_3)$) was determined via the thermodynamic relation (15):

$$\Delta G(*NO_3) = G(*NO_3) - G(HNO_3) + \frac{1}{2}G(H_2) - G(*) + \Delta G_{correct} \tag{15}$$

Here, $G(*NO_3)$, $G(HNO_3)$ and $G(*)$ are the Gibbs free energy of adsorbed nitrate, vaporization of $HNO_3(l)$ and the clean catalyst surface, respectively, while $\Delta G_{correct}$ denotes the adsorption energy correction[60,61].

$$\Delta G_{correct} = \Delta G_{S1} + \Delta G_{S2} \tag{16}$$

$\Delta G_{S1}$ is the Gibbs free energy of formation of $HNO_3(l)$ from $NO_3^-$ (l) (Eq. 12: $\Delta G_{S1} = 0.317$ eV). $\Delta G_{S2}$ is the Gibbs free energy of vaporization of $HNO_3(l)$, which was calculated from the Gibbs free energy difference between the $HNO_3(l)$ and $HNO_3(g)$ (Eq. 13: $\Delta G_{S2} = 0.074$ eV).

### Direct ammonia product recovery in the integrated three-chamber reactor

The coupled electrochemical $NO_3^-$-to-$NH_3$ conversion and in situ nitrate reduction process were accomplished in a three-chamber reactor. The three chambers include an anode chamber ($20 \times 20 \times 10$ mm³), a cathode chamber ($20 \times 20 \times 10$ mm³), and an acid absorption chamber ($20 \times 20 \times 10$ mm³). A three-electrode system was constructed in the flow cell: a piece of Pt as the counter electrode, and the synthesized metamaterial catalysts and Ag/AgCl were used as the working electrode and reference electrode, respectively. The volume of the electrolyte was 500 mL for the anode and cathode chambers and 1 L for the acid absorption chamber. A Nafion@117 cation exchange membrane was used to separate the cathodic and anodic chambers. A piece of commercial gas exchange membrane was applied to separate the cathode chamber and the acid adsorption unit. The anodic and cathodic electrolyte flow rate and dilute HCl (0.3 mM) solution in the acid absorption chamber were set as 60 ml min⁻¹. The adsorption efficiency of $NH_4^+$-N in the acid absorption chamber was calculated using the following equations:

$$A_{NH_4^+-N}(\%) = \frac{acid\ collected\ NH_4^+ - N}{totally\ generated\ NH_4^+ - N} \tag{17}$$

### Mechanical performance tests

As-received commercial Ni foams with a thickness of 5 mm were cut into cubes with an approximate weight and three-dimensional size as the FeCoNi metamaterials. Subsequently, in situ Uniaxial compression experiments of dual-scale shell-lattice metamaterials and Ni foam cubes were conducted inside the SEM chamber at room temperature using a micromechanical tester (Gatan Microtest ™). All samples were performed at a prescribed strain rate of $1.0 \times 10^{-3}$ s⁻¹. At least three specimens for each test were examined for reproducibility. When

there is no apparent first peak strength in the stress-strain curves, the plateau stress is considered the compression strength.

### Calculation of energy consumption

The specific energy consumption (SEC) (kWh Kg⁻¹ N) was calculated according to the following equation:

$$SEC = \frac{E_{cell}It \times 10^{-3}}{m} \tag{18}$$

where $E_{cell}$ is the cell voltage ($V$), $I$ is the electric current intensity ($A$), t is the duration of the reaction (h) and m is the mass of nitrate converted (Kg-N).

## Data availability

The data supporting the findings of this study are available within the article and its supplementary information. Source data are provided with this paper.

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

## Acknowledgements

The research was financially supported by Hong Kong RGC general research fund (No. 11200623, Y. L.) and RGC CRF project (No. C7074-23G, Y. L.); Innovation and Technology Fund (ITP/028/22TP and ITP/058/23TP, X. S.); Changsha Municipal Science and Technology Bureau (kh2201035, Y. L.); National Natural Science Foundation of China (No. 52201176, X. X. L.); Young Talent Support Project·of Guangzhou Association for Science and Technology (Grant No. QT2024–041, X. X. L.). L. Q. W and Y. L. acknowledge the support from the Hong Kong Branch of the National Precious Metals Material Engineering Research Center (NPMM).

## Author contributions

L. Q. W., D. Y., and J.U. S. conceptualized the study. L. Q. W., D. Y., and J.U. S. fabricated the samples. J. H. D. and R. L. analyzed the architecture curvature. L. Q. W., H. L. F., X. X. L., and X. Z. performed the microstructure characterization. D. Y. and M. X. C. conducted the catalysis experiments and analyzed the corresponding data. L. Q. W. and D. Y. wrote the manuscript. D. Y., J. U. S., X. S., J. C. H., and Y. L. reviewed and revised the manuscript. X. S., J. C. H., and Y. L. supervised the project.

## Competing interests

The authors declare no competing interests.
