## [Transparent Peer Review file · Nature Communications]

Harnessing screw dislocations in shell-lattice metamaterials for efficient, stable electrocatalysts

Corresponding Author: Professor Yang Lu

Version 0:

Reviewer comments:

Reviewer #1

(Remarks to the Author)

This manuscript introduces an innovative approach to fabricating FeCoNi metamaterials with TPMS structures, utilizing 3D-printed hydrogels as nitrate precursor impregnation templates, followed by sintering and reduction processes. The primary novelty of this work lies in the use of TPMS lattice structures, where the curved surfaces induce tensile or compressive strains during the growth of FeCoNi crystals, resulting in the formation of additional dislocations. The authors have conducted extensive and impressive microstructural characterizations, greatly enhancing the work's appeal. This manuscript can be accepted after tackling the following questions.

- The manuscript emphasizes that the additional dislocations are attributed to the growth of FeCoNi crystals on non-planar substrates. Could a similar effect be achieved using alternative methods, such as impregnating porous organic foams as templates?

- What is the specific reason for emphasizing the dual-scale lattice design? Why not use a single small-sized TPMS lattice structure instead? According to Figure S1, the larger pores of the hydrogel sample measure approximately 2–3 mm.

Additionally, I am curious about the micro-pore size in the green hydrogel part before sintering, given that the final parts exhibit micro-pores around 100 μm . What is the overall shrinkage rate from the hydrogel sample to the final product?

- As noted in the manuscript, catalytically active nanomaterials are often coated onto porous inert catalyst supports. This study's SEM images of the fabricated samples primarily reveal macro-pores generated through 3D printing. Are the macro-pores controllable?

- The manuscript compares the strength of the prepared samples with commercial Ni foam, demonstrating superior strength for the prepared samples. However, the strength also depends on factors such as material type and porosity. Discussion and clarification are needed.

Reviewer #2

(Remarks to the Author)

This manuscript presents the development of an efficient and stable electrocatalyst for nitrate-to-ammonia conversion. A novel 3D printing method incorporating screw dislocations is used to create a FeCoNi shell-lattice metamaterial catalyst with an ultrahigh dislocation density. This catalyst demonstrates impressive electrochemical properties for nitrate-to-ammonia conversion. Additionally, the researchers also developed an integrated reactor system for continuous ammonia production. Here are some concerns, which should be addressed to strengthen the manuscript and enhance its clarity.

1. The DFT calculations demonstrate the role of step defects and lattice strain in the FeCoNi catalyst in enhancing NO_3^- adsorption and lowering the energy barrier for NO_3RR activity compared to single metal slabs. However, the specific electrocatalytic contributions of Fe, Co, and Ni within the alloy remain unclear. Could the authors elaborate on their rationale for selecting this specific multi-metal composition? Would catalysts with only one or two of these metal components exhibit comparable activity? Further experimental or calculation investigation into the individual and synergistic roles of these elements could provide valuable insights into the catalyst's functionality and optimized design.

2. Figure 3 presents a curvature-dependent increase in dislocation density within the metamaterial catalyst. Does this increase in density predominantly involve screw dislocations, or does it also lead to a higher proportion of edge dislocations? Could the authors provide further insights into the relationship between curvature, dislocation type, and

resulting strain fields within the catalyst structure, and how these different strain fields could influence the observed NO₃RR activity? Additionally, while this work highlights the contribution of screw dislocations and resulting step defects to enhanced NO₃RR activity, it doesn't explicitly address the potential influence of strain induced by edge dislocations. Further investigation into the activity of surface dominated by edge dislocations would provide a more comprehensive understanding of the factors contributing to the enhanced electrocatalytic performance.

3. Please provide the applied potentials for Figure 4b, c, and 6f, and specify the volume of the electrolyte used for the long-term stability test. Additionally, the current density in Figure 4c shows a decrease (absolute value) within each 24-hour electrolysis cycle for the initial 216 hours (9 cycles). However, after 216 hours the current density stabilizes at a roughly constant value, especially in the last few cycles. This change occurs despite the reported "negligible changes in the morphology and composition" before and after the stability experiments (Figures S26-S27). Could the authors please provide an explanation for this phenomenon?

4. In Figure 4a, the current density is normalized by the ECSA. Please clarify whether the current density or yield rate values reported in Figures 4b-d, 6f, and S17, S18, and S20 are normalized by ECSA or geometric area?

5. While the estimated cost of \$3.0 per kg-N is impressively lower than existing methods, this analysis does not consider the cost of treating "nitrate-containing wastewater" to obtain the nitrate source. This makes the cost comparison less persuasive. Additionally, including energy efficiency based on Figure S21 or 6f would allow for a more straightforward evaluation of the process's economic performance.

6. The authors convincingly demonstrate the superior mechanical properties of the dual-scale shell-lattice metamaterial compared to commercial Ni foam using in situ uniaxial compression tests. However, the manuscript would benefit from a more thorough discussion of how these enhanced mechanical properties contribute to improved catalyst performance under practical electrocatalysis conditions. This connection is crucial for establishing the significance of the mechanical tests.

7. In Figure 6h, "interface stability" is listed as a characteristic for comparison between different catalysts, and the screw dislocation-driven metamaterial is shown to have the best performance. However, this is misleading because the screw dislocation-driven metamaterial is monolithic, inherently eliminating the catalyst-support interface. Therefore, "interface stability" is not applicable in this case. To avoid this confusion, please consider replacing "interface stability" with a more appropriate characteristic.

8. The manuscript highlights the advantage of a unified process to avoid heterointerface formation. However, the fabrication process involves multiple steps, including DLP printing of a hydrogel scaffold, immersion in a metal salt precursor, calcination to form metal oxides, and final reduction. Could the authors elaborate on which specific aspects of this multi-step process, and the resulting material characteristics, contribute to achieving an interface-free catalyst?

9. A higher concentration of NaNO₃ (0.5 M) was used for in situ Raman experiments. Could the authors please explain the reason behind using a higher concentration for the in situ Raman study? Furthermore, in the discussion of the in situ Raman results in Supplementary Note 2, it is stated that the antisymmetric bending vibration of adsorbed *NH₃ at 1591 cm⁻¹ disappeared when the potential was decreased to -0.5 V. However, based on Figure S28, this peak appears to become more prominent at -0.5 V. Could the authors please double-check the Raman spectra and revise the description accordingly?

Reviewer #3

(Remarks to the Author)

This paper describes a method for creating advanced catalysts using 3D printing technology. It focuses on screw dislocations to enhance both the manufacturing process and the catalytic performance of the FeCoNi shell-lattice metamaterials. By inducing strain effects, these dislocations improve nitrate-to-ammonia conversion efficiency. The concept of utilizing screw dislocations in 3D printing to enhance catalytic performance is innovative and can bridge the gap between nanoscale material design and macroscale manufacturing. Here are some comments and questions:

1. Could you provide more details on how the screw dislocations are introduced and controlled during the 3D printing process?

2. How does the dislocation density compare to that of conventionally manufactured catalysts?

3. The authors mention that "78% of the dislocations were screw-type." Is it possible to provide more insight into how this percentage was determined from the STEM analysis?

4. The dislocation density was estimated via XRD diffraction peak broadening using the Williamson-Hall method and also measured through EBSD. A clarification on how the two methods compare or validate each other could strengthen the manuscript.

5. It would be interesting to know the proportion of mixed dislocations relative to screw and edge dislocations and whether these contribute differently to the catalytic activity or structural properties.

6. It would be valuable to explain how the lattice strain directly impacts the catalytic activity. For instance, how does a 6.3% lattice tension (as seen in the highest curvature region) influence the material's catalytic efficiency or stability?

7. The comparative analysis between dual-scale shell lattice and reference metamaterials (simple cubic and single-order shell lattice) is valuable for demonstrating the effects of curvature. However, it would be helpful to expand on how these materials' differing mean absolute Gaussian curvatures (MAGC) specifically influence their catalytic properties. Are the increased dislocation densities in the curved metamaterials contributing to higher catalytic efficiency, and if so, how?

8. Given the challenges involved in calculations, how did the authors calculate the energy of the charged NO₃⁻ ion?
9. The elementary reaction steps of NO₃RR described in the Computational Methods section contradicts the Free-energy diagram presented in Figure 5e, which requires careful modification for consistency.
10. Could the authors expand on how dislocations and strain specifically affect other intermediate steps, such as *NO and *NH₂?
11. Did the author consider solvation effect in their calculations? It is quite important factor to be considered in these electrocatalysis

Version 1:

Reviewer comments:

Reviewer #1

(Remarks to the Author)

The authors have addressed the concerns well.

Reviewer #2

(Remarks to the Author)

I sincerely appreciate the authors' efforts in addressing all my comments effectively except one minor unresolved question regarding energy efficiency in comment 2.5. Energy efficiency (EE) for an electrocatalytic process can be calculated using the applied potential and Faradaic efficiency. It is defined as the percentage of energy reserved in the targeted product (here ammonia) divided by the total energy input to the electrolysis system. Could the authors provide the calculated EE value based on either Figure S21 or 6f.

Additionally, in the three-chamber reactor, the authors mentioned Nafion 117 membrane permitted the transport of protons from the anode to the cathode to counteract the OH⁻ produced at the cathode and maintain a constant pH in the cathodic electrolyte. While Nafion's high proton conductivity (up to 0.2 S/cm) allows for proton transport between compartments, I'm concerned that this transport may not be sufficient to fully equilibrate the pH between the two sides during electrolysis under the applied reaction rate. Could the authors clarify this.

Reviewer #3

(Remarks to the Author)

The authors have made visible efforts to address the comments raised by the reviewers, and the revised manuscript reflects some improvements in clarity and organization. Their responses indicate an attempt to clarify the modeling framework and experimental validation, which is appreciated.

However, despite these revisions, I find that the overall depth, and scientific rigor of the work still fall short of the standards expected for publication in this journal. The core contributions remain somewhat incremental, and several aspects of the modeling assumptions, data interpretation, and novelty of the findings are not convincingly developed. In light of these considerations, I believe that the manuscript, in its current form, would be more appropriately directed toward a more specialized or lower-tier journal. I therefore do not recommend publication in this journal.

Version 2:

Reviewer comments:

Reviewer #2

(Remarks to the Author)

The authors have made substantial efforts to effectively address my comments. I recommend accepting the manuscript as is.

Response to Reviewers

Harnessing screw dislocations in shell-lattice metamaterials for efficient, stable electrocatalysts

Liqiang Wang[†], Di Yin[†], James Utama Surjadi[†], Junhao Ding, Huangliu Fu, Xin Zhou, Rui Li, Mengxue Chen, Xinxin Li, Xu Song^{*}, Johnny C. Ho^{*}, Yang Lu^{*}

We sincerely appreciate the comprehensive comments and constructive suggestions raised by reviewers. Based on that, we have revised our manuscript accordingly. We believe the manuscript has been improved. In the following context, we have prepared a response to the comments in a point-by-point fashion, detailed below (red: comments from the reviewer; black: response; blue: revised manuscript)

Reviewer #1

General comment:

This manuscript introduces an innovative approach to fabricating FeCoNi metamaterials with TPMS structures, utilizing 3D-printed hydrogels as nitrate precursor impregnation templates, followed by sintering and reduction processes. The primary novelty of this work lies in the use of TPMS lattice structures, where the curved surfaces induce tensile or compressive strains during the growth of FeCoNi crystals, resulting in the formation of additional dislocations. The authors have conducted extensive and impressive microstructural characterizations, greatly enhancing the work's appeal. This manuscript can be accepted after tackling the following questions.

Response: We sincerely thank the reviewer for his/her positive feedback on our manuscript. We are particularly appreciative of their recognition of the concept of screw dislocation-driven synthesis of TPMS metamaterial catalysts. The formation of curvature-induced dislocations and interface-free surface nanostructure in FeCoNi TPMS metamaterials has been demonstrated to significantly enhance nitrate reduction catalytic activity and stability. We appreciate the constructive suggestions provided, which will help us improve the quality of our work. Below, we have addressed each of the reviewer's specific comments in detail. We believe these revisions will significantly enhance the clarity, rigor, and overall impact of our manuscript, making it more suitable for publication in *Nature Communications*.

Comment 1.1:

The manuscript emphasizes that the additional dislocations are attributed to the growth of FeCoNi crystals on non-planar substrates. Could a similar effect be achieved using alternative methods, such as impregnating porous organic foams as templates?

Response: Thanks. Although we do not specialize in the field of porous organic foams, we have conducted a literature review to explore the potential of using porous organic foam templates to

achieve a similar effect. Such as in a very recent paper, Wang et. al [R1.1] encapsulated Fe acetate in the cage of imidazole framework-8 (ZIF-8) metal-organic frameworks (MOFs) via a general host-guest strategy, obtaining Fe-ZIF-8 dodecahedra precursor. A thin layer of mesoporous silica was subsequently coated onto the surface of Fe-ZIF-8. During pyrolysis, the uniform SiO₂ coating layer can generate an outward adsorption force, thus the dodecahedron planar surfaces collapse, leaving a concave morphology after removing the SiO₂ shell. It can produce a sample with single Fe atoms anchored on a highly curved carbon substrate. They discovered that the curvature of the carbon substrate can reduce the bond length of Fe-N, thereby shifting the d-band center downward to achieve optimal adsorption for oxygenated species. We consider that this is similar to our substrate curvature effect, which relates to strain engineering in surface catalysts. In their case, since the single Fe atom has no spatial capacity to generate dislocation defects, it directly results in lattice strain behavior. In our work, dislocation multiplication in curved space serves as the carrier of increased elastic strain energy, resulting in the controllable formation of twisted nanosteps and significant lattice strain.

Besides, we also try to emphasize that our work highlights a screw dislocation-driven 3D printing strategy. This approach demonstrates multiple significant advantages over most conventional random porous organic foams used as templates, such as the fabrication of monolithic metamaterial catalysts with highly customizable topological curvature, interface-free surface nanomaterials, and enhanced mechanical robustness due to the engineered 3D architecture.

Comment 1.2:

What is the specific reason for emphasizing the dual-scale lattice design? Why not use a single small-sized TPMS lattice structure instead? According to Figure S1, the larger pores of the hydrogel sample measure approximately 2–3 mm. Additionally, I am curious about the micro-pore size in the green hydrogel part before sintering, given that the final parts exhibit micro-pores around 100 μm. What is the overall shrinkage rate from the hydrogel sample to the final product?

Response: We appreciate this valuable comments from the reviewer. In this work, to investigate the influence of curvature-induced dislocation multiplication on electrocatalytic activity, we aim to design three types of metamaterials with significantly increased mean absolute Gaussian curvatures. However, given the potential influence of relative density and specific surface area in catalytic performance, it is essential to maintain consistent relative density and specific surface area across the three distinct metamaterials. As suggested by the reviewer, while using smaller unit cell size for single-order lattice would indeed result in higher mean absolute Gaussian curvature, it would require an increase in specific surface area. Because single-order lattice always exhibited a positive correlation between mean absolute Gaussian curvatures and specific surface area, as shown in Fig. R1 and Table R1. Furthermore, decreasing the unit size necessitates a reduction in wall thickness to maintain consistent relative density, which makes it more challenging to fabricate due to the limited resolution of the 3D printing technique. In contrast, at any given relative density and specific surface area, dual-scale lattice achieves drastically higher mean absolute curvatures (more than an order of magnitude difference) compared to single-order lattices.

To highlight this point, we slightly revised corresponding sentences in different positions.

“Screw dislocation-mediated integrated manufacturing of metamaterial catalysts” section:

“At any given relative density and specific surface area, dual-scale TPMS design strategy can achieve drastically higher mean absolute curvatures compared to single-order lattices.”

“Dislocation multiplication in curved space” section:

“To enhance comparability, they maintained consistent relative densities of $\sim 7\%$ and specific surface areas of $\sim 2.5 \text{ mm}^{-1}$, while achieving significantly different mean absolute Gaussian curvatures (measured at 0, 192, and 4650 mm^{-2} , denoted as S0, S192, and S4650, respectively, as shown in Fig. 3d).”

Fig. R1 The relationship between mean absolute Gaussian curvature and specific surface area in single-order gyroid lattices (assuming overall sample size maintained at a 5 mm cubic dimension).

Table R1. The relationship among unit size, specific surface area, and mean absolute Gaussian curvature in single-order gyroid lattices (assuming overall sample size maintained at a 5 mm cubic dimension).

Unit size (mm)	Surface area (mm ²)	Specific surface area (mm ⁻¹)	Mean absolute Gaussian curvatures
5	77.3	0.6184	0.69
4.5	83.3	0.6664	0.79
4	99.41	0.79528	1.03
3.5	111.5	0.892	1.41
3	125.99	1.00792	1.83

2.5	154.62	1.23696	2.77
2	193.3	1.5464	4.29
1.5	262.65	2.1012	7.66
1	386.7	3.0936	17.42

Regarding the second question concerning the micro-pore size in the green hydrogel before sintering, we note that previous Fig. S1 does not include the green part. We assume that the reviewer is referring to the yellow metal-salt-rich architecture before sintering. As the metal-salt-rich hydrogel metamaterial was preliminarily dried in a vacuum oven at 50°C for 48 hours (**Methods in manuscript**), a rapid shrinkage occurred, resulting in a deviation from the original 3D model design. The micro-pore size was measured using optical microscopy, revealing an average diameter of 294 μm , corresponds to the diameter of 414 μm in as-printed scaffolds (**Table S1**).

Subsequently, we calculated the overall shrinkage rate throughout the process, including linear shrinkage and mass loss (**Fig. S2**). Linear shrinkage was calculated from lattice side length from the as-printed hydrogel state to the reduced alloy state, measured from optical or SEM images. Mass loss was calculated from dried metal-salt-rich state to the reduced alloy state. **Table S1** details the changes in feature size at each hierarchical level between the as-printed hydrogel metamaterial and the final FeCoNi metamaterial architecture. **Fig. R2** provides a schematic diagram illustrating the measurement approach for each architectural hierarchy. After calculation, we found each hierarchy undergoing approximately 72.1% linear shrinkage, accompanied by an estimated 92.8% mass loss of sample during calcination process.

Fig. S2 highlight geometry evolution during the fabrication process. The detailed measurement result can be found in **Table S1**. The related text was added to the “*Screw dislocation-mediated integrated manufacturing of metamaterial catalysts*” section.

“We observed a uniform linear shrinkage at each hierarchical level of approximately 72.1%, accompanied by an estimated 92.8% mass loss of sample during the calcination process (**Fig. S2** and **Table S1**)”

Fig. S2 Geometry morphology and feature size evolution during the fabrication process

Comment 1.3:

As noted in the manuscript, catalytically active nanomaterials are often coated onto porous inert catalyst supports. This study's SEM images of the fabricated samples primarily reveal macro-pores generated through 3D printing. Are the macro-pores controllable?

Response: In our work, the macro-pores generated via 3D printing are indeed highly controllable. As our study in *comment 1.2*, although a remarkable linear shrinkage can be observed during the drying and sintering process, the shrinkage process is isotropic and linear shrinkage rate at each hierarchy maintains consistent around 72% (Table S1 and Fig. R2). Therefore, we can achieve stable geometric morphology in the final architecture with minimal distortion, as shown in Fig. S2. By accounting for the linear shrinkage in advance, we can still design and achieve the desired lattice pore size in the final alloy components. In addition, in terms of geometric freedom, the 3D printing model developed in this study, the dual-scale shell lattice, exhibits a hierarchical porous network with distinct three-dimensional features spanning multiple orders of magnitude, ranging from tens of microns to several

centimeters, highlighting our exceptional controllability over lattice structure.

Table S1 Linear shrinkage rate during the fabrication process

Length at each hierarchy	As-printed hydrogel metamaterial	Final FeCoNi metamaterial	Shrinkage rate (%)
Whole sample (mm)	15.1	4.2	72.2
First order pore (mm)	3.1	0.9	71.0
Second order pore (μm)	414.2	117.6	71.6
Minimum wall thickness (μm)	76.5	10.2	73.6
Average	-	-	72.1

Fig. R2 Schematic diagram of length measurement at each hierarchy

Comment 1.4:

The manuscript compares the strength of the prepared samples with commercial Ni foam, demonstrating superior strength for the prepared samples. However, the strength also depends on factors such as material type and porosity. Discussion and clarification are needed.

Response: We highly appreciated the insightful and valuable comment here. Indeed, the mechanical properties of metamaterials were typically influenced by their material composition and geometry features. We first clarify that Porosity = $(1 - \text{Relative density}) \times 100\%$. Relative density, defined as the ratio of the density of the lattice structure to the density of the solid material from which it is made, directly influencing the compression strength and energy absorption of metamaterials. In this work, we have seriously considered this point. Due to the inability to freely adjust the porosity of commercial Ni foam, we elaborately designed the FeCoNi dual-scale gyroid to match the relative density of $\sim 7\%$ found in commercial Ni foam, ensuring that all uniaxial compression experiments were conducted under consistent relative density conditions. We slightly revised related sentence in the manuscript to highlight the consistent relative density.

“To address this, it is necessary to further investigate the mechanical properties of our optimized dual-scale gyroid, especially compared to commercial Ni foam (a commonly used support material) at an equivalent relative density of $\sim 7.0\%$.”

Second, to separately investigate the influence of material and architecture on mechanical properties, we fabricated a new reference sample of pure Ni dual-scale gyroid with a consistent relative density of approximately 7%. The mechanical performance of the pure Ni dual-scale gyroid was evaluated through uniaxial compression experiments under the same testing conditions. The comparison of stress-strain curves between Ni dual-scale gyroid and Ni foam proves that our architectural design played an important role in enhanced mechanical strength (Fig. S40a). When we further studied the influence from material design, the FeCoNi dual-scale gyroid exhibited enhanced compressive strength and specific strength (over 2 times) compared to the pure Ni dual-scale gyroid. From SEM image, the fracture surface morphologies of the samples subjected to FeCoNi metamaterials displayed numerous dimples, indicating a ductile fracture mode. Conversely, quasi-cleavage fracture morphology was observed at Ni metamaterials, demonstrating the brittle fracture mode (Fig. S40b). This demonstrates that our multicomponent composition design can effectively enhance the strength and ductility. The energy absorption and specific energy absorption, calculated from the area under the corresponding stress-strain curves [R1.2], revealed an over 50% increase in the FeCoNi compared to the pure Ni sample with the same architecture (dual-scale gyroid). Previous studies found that, enhanced mechanical properties in FeCoNi alloy may be attributed to the severe lattice distortion [R1.3] and the sluggish diffusion effect in multicomponent alloys [R1.4].

Overall, the FeCoNi dual-scale gyroid achieved an outstanding balance of mechanical properties, which can be attributed to integrating the advantage of shell-based architecture and multicomponent composition design. Therefore, the figure and manuscript has been revised to include additional discussion regarding both material and architectural aspects. Due to space constraints in the main text, a separate and more detailed discussion has been provided in *Supplementary Note 5*.

“*Practical applications of metamaterial catalysts*” section

“Integrating the advantage of architecture and materials (Supplementary Note 5 and Fig. S39-41), Our FeCoNi dual-scale gyroid not only demonstrates significantly higher strength compared to Ni foam, with a compression strength 3 times higher (Fig. 6a), but also exhibit a more homogeneous deformation.”

“In addition to the contributions from architectural design, the optimization of the multicomponent composition also played a significant role in enhancing both strength and ductility. With the same dual-scale gyroid design, FeCoNi sample still exhibited 2 times higher strength than pure Ni and displayed ductile fracture surface morphologies with numerous dimples (Fig. S40). Overall, FeCoNi dual-scale gyroid achieved an outstanding balance of mechanical properties by integrating the advantage of shell-based architecture and multicomponent composition design, outperforming most of the previous reported foam structures or lattice structures (Fig. 6c-d)”

“*Supplementary Note 5*” section

Materials design advantages

The stress-strain curve in Fig. S40, the FeCoNi dual-scale gyroid exhibited enhanced compressive strength and specific strength (over 2 times) compared to pure Ni dual-scale gyroid. We further observed the fracture microstructure using SEM image. The fracture surface morphologies of the samples subjected to FeCoNi metamaterials displayed numerous dimples, indicating a ductile fracture mode. Conversely, quasi-cleavage fracture morphology was observed at Ni metamaterials, demonstrating the brittle fracture mode (Fig. S40). This demonstrates that our multicomponent composition design effectively enhances the strength and ductility of the dual-scale gyroid metamaterials. The energy absorption and specific energy absorption, calculated from the area under the corresponding stress-strain curves, revealed an over 50% increase in the FeCoNi dual-scale gyroid

compared to the pure Ni dual-scale gyroid. Previous studies found that, enhanced mechanical properties in FeCoNi alloy may be attributed to the severe lattice distortion and the sluggish diffusion effect in multicomponent alloys.

Fig. S40 Mechanical properties and deformation microstructure of FeCoNi dual-scale gyroid, Ni dual-scale gyroid, and Ni foam. (a) Compression stress-strain curve of different samples with an equivalent relative density of ~7%. (b) Comparison of fracture surface morphologies. (c) Comparison of compression strength/ specific compression strength and energy absorption/ specific energy absorption.

In addition, with the inclusion of mechanical properties data for the Ni dual-scale gyroid, the Ashby maps depicting specific compression strength versus density and energy absorption per unit volume versus density also were updated. The revised figures can be shown below.

Fig. 6 (c) Ashby map of specific compression strength versus density and (d) energy absorption per unit volume versus density of shell-lattice metamaterials compared to reported metamaterials⁴⁰⁻⁴⁸.

References

- [R1.1] Wang Q, Lyu L, Hu X, et al. Tailoring the Surface Curvature of the Supporting Carbon to Tune the d-Band Center of Fe-N-C Single-Atom Catalysts for Zinc-Urea-Air Batteries[J]. *Angewandte Chemie International Edition*, e202422920.
- [R1.2] Wang, Liqiang, et al. "High-precision Cu alloy microlattices with superior energy absorption capacity enabled by nanoprecipitation engineering." *Scripta Materialia* 239 (2024): 115801.
- [R1.3] He Q, Yang Y. On lattice distortion in high entropy alloys[J]. *Frontiers in Materials*, 2018, 5: 42.
- [R1.4] Jin K, Zhang C, Zhang F, et al. Influence of compositional complexity on interdiffusion in Ni-containing concentrated solid-solution alloys[J]. *Materials Research Letters*, 2018, 6(5): 293-299.

Reviewer #2

General comment:

This manuscript presents the development of an efficient and stable electrocatalyst for nitrate-to-ammonia conversion. A novel 3D printing method incorporating screw dislocations is used to create a FeCoNi shell-lattice metamaterial catalyst with an ultrahigh dislocation density. This catalyst demonstrates impressive electrochemical properties for nitrate-to-ammonia conversion. Additionally, the researchers also developed an integrated reactor system for continuous ammonia production. Here are some concerns, which should be addressed to strengthen the manuscript and enhance its clarity.

Response: We are deeply grateful to the reviewer for his/her positive feedback on our work, which is very important for us to further improve our work and the quality of the manuscript. Enclosed below are our detailed point-to-point responses to your suggestions.

Comment 2.1

The DFT calculations demonstrate the role of step defects and lattice strain in the FeCoNi catalyst in enhancing NO₃⁻ adsorption and lowering the energy barrier for NO₃RR activity compared to single metal slabs. However, the specific electrocatalytic contributions of Fe, Co, and Ni within the alloy remain unclear. Could the authors elaborate on their rationale for selecting this specific multi-metal composition? Would catalysts with only one or two of these metal components exhibit comparable activity? Further experimental or calculation investigation into the individual and synergistic roles of these elements could provide valuable insights into the catalyst's functionality and optimized design.

Response: Thank you for your insightful questions regarding the rationale of using FeCoNi multi-metal composition and the roles of its individual components. Earth-abundant transition metals (Fe, Co, Ni) have been widely recognized as promising candidates for NO₃⁻ reduction [R2.1-R2.3]. Inspired by their potential, we selected these elements to design FeCoNi medium-entropy alloy (MEA) compositions. As suggested by the reviewer, single and binary metal components should be used to determine whether a simpler multi-metal system exists for efficient NO₃RR. In that way, the performance of those metal systems as electrocatalysts for NO₃RR was studied. We further printed a series of dual-scale shell metamaterial catalysts with different composition components using the same process and geometry parameters, including Fe, Co, Ni, FeCo, FeNi, and CoNi (Fig. S22a). To assess their catalytic performance, one-hour electrolysis tests were conducted under -0.7 V applied potential in a 0.5 M Na₂SO₄ electrolyte containing 0.1 M NaNO₃. Amperometric (i-t) analyses revealed that the FeCoNi MEA achieved markedly higher NH₃ Faradaic efficiency and production rates than monometallic and binary systems (Fig. S22b).

Therefore, the related texts were added to the main text, and the figure was added as Supplementary Information, as shown below.

“Preparation of monolithic metamaterials catalyst” in Methods section.

“To conduct the comparison experiment, we further fabricated a series of dual-scale shell metamaterial catalysts with different composition components using the same process and geometry parameters, including Fe, Co, Ni, FeCo, FeNi, and CoNi.”

“Electrocatalysis performance and mechanism” section.

“Amperometric (i-t) analyses revealed that the FeCoNi medium-entropy alloy (MEA) achieved markedly higher NH_3 Faradaic efficiency and production rates compared to monometallic and binary systems (Fig. S22).”

Fig. S22 (a) The fabrication of reference samples with different compositional elements. (b) FE_{NH_3} and Y_{NH_3} of Fe, Co, Ni, FeCo, FeNi, and CoNi, and FeCoNi metamaterial catalysts.

To elucidate the correlation between the synergistic effects induced by multi-elemental mixing in MEA and their electrocatalytic performance, we conducted density functional theory (DFT) calculations. We first analyzed charge density distributions for surface atoms in both pure metallic systems and the MEA to examine how multiple elements influence local electronic structure at atomic scales (Fig. S23a). Notably, the FeCoNi MEA displayed substantial charge redistribution relative to single-element metallic structures. In particular, electron density values for Fe atoms in the MEA were reduced relative to their pure metal states, whereas Co and Ni demonstrated increased localized electron concentrations within the alloy matrix. This redistribution pattern indicates a directional electron transfer process from Fe constituents to Co/Ni elements, aligning with established research findings [R2.4-R2.6]. Moreover, Bader charge analysis was employed to quantify charge variations for surface atoms within the FeCoNi MEA (Fig. S23b). While all metallic constituents displayed consistent trends in electronic structure modifications, the magnitude of charge transfer varied significantly across elements, influenced by

their distinct coordination environments [R2.7]. Fe atoms exhibited a charge gain ranging from 0.05 to 0.14 |e|, whereas Co and Ni atoms experienced a charge loss from 0.001 to 0.05 |e| and 0.04 to 0.12 |e|, respectively.

Partial projected density of states (PDOS) analysis further confirmed significant electron redistribution in the multi-component system (Fig. S23c). Fe sites dominate near the Fermi level (EF), while Co and Ni occupy deeper energy levels. The 3d-orbitals of these elements form a near-continuous distribution via d-d hybridization, enabling efficient electron transfer in the FeCoNi MEA [R2.8]. Element-specific PDOS and d-band center models were analyzed to explore adsorbate-alloy electronic interactions (Fig. S23c)[R2.9]. The d-band centers (ϵ_d) of pure Fe, Co, and Ni relative to EF were calculated as -1.04 , -1.12 , and -1.20 eV, respectively, indicating stronger adsorption by Fe. In the MEA, Fe ($\epsilon_d = -0.87$ eV) exhibits d-states closer to EF, enhancing NO_3RR intermediate adsorption. Conversely, Co ($\epsilon_d = -1.14$ eV), and Ni ($\epsilon_d = -1.34$ eV) show downshifted d-band centers, reducing adsorption strengths [R2.10]. These electronic variations create a broad adsorption energy landscape in the MEA, ideal for multi-step NO_3RR catalysis. The system's ability to accommodate diverse binding energies makes it highly promising for complex catalytic applications.

Fig. S23. (a) Electron density of the FeCoNi MEA. (b) Bader charge analysis of the FeCoNi MEA. (c) A comparison of computed PDOS of each element in the MEA and pure metals (Fe, Co, and Ni).

Theoretical calculations are further applied to illustrate the reactivity properties of multiple active sites in MEA for NO_3RR . Thermodynamically, strong affinity for NO_3^- is a critical prerequisite for efficient NO_3^- -to- NH_3 conversion [R2.11]. The adsorption of bridge-bidentate $^*\text{NO}_3$ on 9 unique coordination environments of FeCoNi catalyst was studied (Fig. 5b and Fig. S32-35), presenting more active sites than the single metal slab. The nine adsorption sites of the FeCoNi-LBL slab exhibit a wide range of $\Delta G^*_{\text{NO}_3}$ values, spanning from -0.01 to -1.26 eV. However, adsorption energy scaling in multi-step NO_3RR restricts using $^*\text{NO}_3^-$ adsorption as a full activity metric for the conversion process. Bayesian chemisorption analyses reveal linear $^*\text{NO}_3^-$ - $^*\text{N}$ adsorption energy correlations in metals, establishing these as key reactivity descriptors [R2.11-R2.12]. Monometallic systems show strong $^*\text{NO}_3^-$ - $^*\text{N}$

scaling (slope = 1.41, $R^2 = 0.96$), matching the theoretical predictions of 1.53 (Fig. S24). MEAs, however, host heteronuclear sites (e.g., Co-Fe, Co-Ni) that disrupt scaling laws. Simulations indicate Fe sites in MEAs stabilize *NO_3 via higher d-band centers, while Co/Ni sites with lower d-band positions cause repulsive interactions. This decouples *NO_3 and *N binding energy tuning, overcoming scaling limitations. This multi-component system, characterized by an expanded adsorption energy range, achieves an optimal balance between intermediate adsorption and desorption, thus overcoming scaling constraints to enable efficient NH_3 electrosynthesis.

Fig. S24 DFT-calculated adsorption energies of *NO_3 and *N on pure metal (red ball) and various binding active sites in the MEA (Fe: blue; Co: yellow; Ni: red).

Overall, multi-element synergy in MEAs optimizes the electrochemical nitrate reduction process by balancing adsorption and reaction dynamics. Due to space constraints in the main text, the discussion above has been entirely put into *Supplementary Note 3*. In addition, we have incorporated a few related sentences into the main text.

“*Electrocatalysis performance and mechanism*” section.

“Moreover, we conducted density functional theory (DFT) calculations to further confirm the synergistic effects induced by multi-elemental mixing in MEA optimize the electrochemical nitrate reduction process (*Supplementary Note 3* and Fig. S23-24).”

Comment 2.2

Figure 3 presents a curvature-dependent increase in dislocation density within the metamaterial catalyst. Does this increase in density predominantly involve screw dislocations, or does it also lead to a higher proportion of edge dislocations? Could the authors provide further insights into the relationship between curvature, dislocation type, and resulting strain fields within the catalyst structure, and how these different strain fields could influence the observed NO_3RR activity? Additionally, while this work highlights the contribution of screw dislocations and resulting step defects to enhanced NO_3RR activity, it doesn't explicitly address the potential influence of strain induced by edge dislocations. Further investigation into the activity of surface dominated by edge dislocations would provide a more comprehensive understanding of the factors contributing to enhanced electrocatalytic performance.

Response: Based on the reviewer's suggestion, we additionally analyzed the influence of curvature on dislocation types using TEM images. Since statistical analysis requires more specimens, we replaced the corresponding TEM image in Fig. 3b with a lower magnification image. All TEM or STEM images were acquired under two-beam conditions to enhance the contrast of dislocations. Due to the significant difference in dislocation density between the small and large curvature areas, when calculating the dislocation type percentage, the small curvature area was zoomed out to increase the number of dislocation lines, while the large curvature area was zoomed in to reduce the number of dislocation lines. The identified dislocation line is compared with the Burgers vector direction. Then the dislocation type is determined if the difference is <10 degrees. If the dislocation line contains two characteristics, the one with the closest match to the dislocation line is chosen. The identified dislocation types were marked on the TEM images using different colored lines (edge: green lines, screw: white lines). (Fig. S14a). We observed that an increase in substrate curvature leads to a significant rise in dislocation density. Additionally, the percentage of edge dislocations gradually increases from 22% to 38% as the substrate curvature increases (Fig. S14b). Although screw dislocations still dominate in areas of large curvature, the increased percentage of edge dislocations in these regions cannot be overlooked. In fact, on small curvature substrates, the elastic strain energy mainly results from the screw dislocation growth mode, thus ideal screw dislocation preserves the orientation of each layer and generates an aligned spiral shape of nanoplates, as shown in Fig. S13. Therefore, screw dislocations constitute a large proportion in areas of small curvature. While on large curvature substrates, an additional elastic energy will be introduced, increasing remarkably with the curvature size. In this case, the generation of both screw dislocations and edge dislocations becomes equally favorable in reducing the system energy, leading to increased percentage of edge dislocation. Regarding the lattice fields of screw and edge dislocations, we have observed and discussed them in Fig. 2f. In the following density functional theory calculations, we considered the distinct atomic structure models of different dislocation defects.

Accordingly, we added some related text in “*Dislocation multiplication in curved space*” section, as shown below.

“Regarding the type of dislocations, the statistical analysis results (Fig. S14a-b) reveal that the percentage of edge dislocations gradually increases from 22% to 38% as the substrate curvature increases. Nevertheless, screw dislocations remain the dominant type even in areas with large curvature.”

Fig. S14 The influence of substrate curvature on dislocation density, dislocation types, and lattice strain.

Figure 3. Dislocation multiplication in curved space. (b) Influence of substrate Gaussian curvature on the dislocation density and lattice strain at surface nanosteps. Scale bar: 100 nm for STEM images.

Then, we address the comment regarding the influence of edge dislocations on catalytic activity. Screw dislocations induce spiral growth patterns (the surface step defects), which enhance the electrochemical active area, and advancing the progress of NO_3^- reduction with a lower thermodynamic barrier of 0.51 eV as compared to that of none stepped FeCoNi-LBL (0.89 eV). Although computational results indicate that screw dislocations dominate in this catalyst, as noted by the reviewer, edge dislocations also play a non-negligible role in modulating catalytic activity. Therefore, we employed density functional theory (DFT) to investigate the structure of edge dislocations and their impact on catalysis (Fig. S38). As a result, the potential-determining step (PDS) on the FeCoNi slab with edge dislocations is the $^*\text{NH}_2\text{OH}$ to $^*\text{NH}_2$ step, with a significantly lower thermodynamic barrier of 0.65 eV. It means that this exposed step sites created by edge dislocations reduce the energy barrier of NO_3RR as compared to FeCoNi-LBL. Moreover, severe strain effects induced by screw and edge dislocation enhance intrinsic catalytic activity by promoting NO_3^- adsorption and lowering the energy barrier of NO_3^- -to- NH_3 conversion. In this way, the coupled effect of these two types of dislocations contributes to the enhanced catalytic activity. Due to the unique role of screw dislocations in generating surface nanosteps and distorted lattice fields, our focus remains primarily on understanding the effects of screw dislocations. While related description of edge dislocation was added in “*Electrocatalysis performance and mechanism*” section, and figure was included as supplementary information.

“Although screw dislocations remain the predominant type across substrates with varying curvatures,

the role of edge dislocations should also be studied in modulating catalytic activity. According to the density functional theory calculation of edge dislocations (Fig. S38), the potential-determining step on the FeCoNi slab with edge dislocations is the $*\text{NH}_2\text{OH}$ to $*\text{NH}_2$ step, with a significantly lower thermodynamic barrier of 0.65 eV. It means that this exposed sites created by edge dislocations reduce the energy barrier of NO_3RR as compared to FeCoNi-LBL.”

Fig. S38 (a) The adsorption models of various intermediates generated during NO_3RR pathways on the surface of FeCoNi slab edge dislocations. (b) Free-energy diagram for NO_3RR on FeCoNi slab edge dislocations.

Comment 2.3

Please provide the applied potentials for Figure 4b, c, and 6f, and specify the volume of the electrolyte used for the long-term stability test. Additionally, the current density in Figure 4c shows a decrease (absolute value) within each 24-hour electrolysis cycle for the initial 216 hours (9 cycles). However, after 216 hours the current density stabilizes at a roughly constant value, especially in the last few cycles. This change occurs despite the reported "negligible changes in the morphology and

composition" before and after the stability experiments (Figures S26-S27). Could the authors please provide an explanation for this phenomenon?

Response: Thank you for your thoughtful comments and questions. Below, we address each point here in detail:

Applied potential:

Figures 4b, 4c: The experiments in these figures were conducted at a fixed potential of -0.7 V vs. RHE. This potential was selected based on prior study of the NO_3^- -to- NH_3 conversion across a broad potential window from -0.1 to -1.1 V in Fig. S21.

Figures 6f: The long-term stability test in this integrated three-chamber reactor was conducted at a fixed potential of -1.0 V vs. RHE. This potential was chosen following the optimization to achieve an optimal trade-off between catalytic activity and operational stability

Electrolyte Volume:

The electrolyte volume for all electrochemical measurements were conducted in a typical H-type cell with 40 mL of electrolyte in each side. The coupled electrochemical NO_3^- -to- NH_3 conversion and in situ ammonia recovery process were accomplished in a three-chamber reactor. The three chambers include an anode chamber ($20 \times 20 \times 10$ mm³), a cathode chamber ($20 \times 20 \times 10$ mm³), and an acid absorption chamber ($20 \times 20 \times 10$ mm³). The volume of the electrolyte was 500 mL for anode and chamber and 1 L for acid absorption chamber.

The updated results are presented in *Methods* of the revised manuscript.

“Amperometric (i-t) analyses and the long-term stability tests were performed at a fixed potential of -0.7 V vs. RHE.”

“The volume of the electrolyte was 500 mL for anode and cathode chamber and 1 L for acid absorption chamber.”

Catalytic stability:

In the initial version of the manuscript, XRD and SEM results (now Fig. S29) exhibited negligible changes in morphology and composition. To further explore potential explanations for the change in current density, we carried out TEM analysis on the sample following the catalytic stability test. The TEM samples were lifted out in medium curvature region. In Fig. S30, we observed significant dislocation tangled with each other in a network way before stability test. After catalytic stability test, the dislocations progressively untangle and straighten out. Although significant changes in dislocation morphology are observed, the dislocation density remains relatively stable. Previous studies revealed that introducing the electric field can decrease the glide barriers of dislocations, thus leading to the dislocation movement [R2.13]. Such “disentanglement” process of dislocations can contribute to the stable release of strain energy [R2.14], relieving part of high energy strain. As our discussion in main text, due to the coupled effect between screw dislocation growth strain energy and curvature-induced additional elastic energy, accumulated strain energy in original samples surpasses a critical threshold, leading to the significant dislocation multiplication behavior with inhomogeneous lattice strain fields. The dislocation movement can effectively achieve strain relaxation during continuous catalytic

reactions, leading to the decrease in current density during the initial 9 cycles. Following the relaxation of high-energy lattice strain, the inherent dislocation defects with stable density serve as the primary contributor to the consistent current density after 216 hours. Similar strain relaxation phenomenon has also been observed in conventional strained catalysts [R2.15].

According to the discussion above, we added related sentences in the main text and the corresponding figure was added as supplementary information, as shown below.

“While slight fluctuations in current density observed during the cycling tests can be attributed to the "disentanglement" process of dislocations (Fig. S30), leading to the typical strain relaxation phenomenon.”

Fig. S30 The comparison of dislocation morphology before and after catalytic stability test. Scale bar: 100 nm.

Comment 2.4

In Figure 4a, the current density is normalized by the ECSA. Please clarify whether the current density or yield rate values reported in Figures 4b-d, 6f, and S17, S18, and S20 are normalized by ECSA or geometric area?

Response: Thanks for highlighting the need for clarity regarding the normalization method. We

specify the normalization approach for each figure. The current density values in Fig. 4c, 6f and S20 are normalized by the geometric area of the electrode. This aligns with standard reporting practices for practical catalytic performance and facilitates direct comparison with literature [R2.16-R2.17]. The NH₃ yield rates in Fig. 4b and 4d are calculated based on geometric area, as reflecting macroscopic reactor output rather than intrinsic activity [R2.18]. The current density in Fig. S18 is normalized by the geometric area of the electrode. The related current density normalized by the ECSA was Figure 4a. The current density in Fig. S17 is normalized by the geometric area of the electrode. The related current density normalized by the ECSA was added in Fig. S17b. We have revised the Fig. S17.

Fig. S17 (a) LSV curves and (b) ECSA normalized partial current densities in LSV curves of FeCoNi-S4650 in 0.5 M Na₂SO₄ electrolyte with and without 0.1 M NO₃⁻-N upon a scan rate of 5 mV s⁻¹.

All explicit statements of the structured response in the captions of Fig. 4b-d, 6f, and S17, S18, S20 are added in *Methods* of the revised manuscript.

“Unless otherwise stated, the current density and yield rate are normalized by the geometric areas of the electrodes.”

Comment 2.5

While the estimated cost of \$3.0 per kg-N is impressively lower than existing methods, this analysis does not consider the cost of treating “nitrate-containing wastewater” to obtain the nitrate source. This makes the cost comparison less persuasive. Additionally, including energy efficiency based on Figure S21 or 6f would allow for a more straightforward evaluation of the process's economic performance.

Response: Thank you for raising these critical points, which would strengthen the robustness of our economic analysis. We acknowledge that the initial cost estimate of \$3.0 per kg-N focused solely on the electrocatalytic NO₃⁻-to-NH₃ conversion process and did not account for upstream nitrate source procurement. As suggested, we have integrated energy efficiency data from Fig. S21 (system-level energy consumption) and Fig. 6f (long-term stability) into the economic evaluation. This integrated three-chamber reactor system assembled with electrochemical NO₃⁻-to-NH₃ conversion and ammonia recovery process exhibits high NH₃ Faraday efficiency (90%) and NH₄Cl recovery (91.3%) with a low electricity cost of 25.6 kWh kg⁻¹ N.

We appreciate your feedback and have deleted the related sentence in “*Practical applications of metamaterial catalysts*” section, as shown below.

This suggests that, if the electricity costs 0.12 dollars per kWh, the removal cost per kg-N would be \$3.0, which is significantly cheaper than that of existing nitrification/denitrification technologies

(\$15.6 per kg-N⁴⁹).

Comment 2.6

The authors convincingly demonstrate the superior mechanical properties of the dual-scale shell-lattice metamaterial compared to commercial Ni foam using *in situ* uniaxial compression tests. However, the manuscript would benefit from a more thorough discussion of how these enhanced mechanical properties contribute to improved catalyst performance under practical electrocatalysis conditions. This connection is crucial for establishing the significance of the mechanical tests.

Response: Mechanical stability is always a key indicator to evaluate the potential industrial applicability of any electrocatalysts, however, existing research efforts appear disproportionately focused on functional performance optimization, with insufficient attention given to this essential fundamental aspect. Especially, the harsh industrial operating conditions, including high current densities, elevated temperatures, and mechanical stresses, impose more rigorous demands on the mechanical stability of catalysts [R2.19]. Conventionally, the mechanical stability of electrocatalysts mainly involves the interface adhesion strength between support and nanomaterials and the intrinsic strength of porous catalyst support. On the one hand, the impact and drag forces induced by electrolyte convection and turbulence can cause catalyst detachment, which is a common problem especially for conventional powdery catalysts [R2.20]. Our screw dislocation-mediated 3D printing of monolithic metamaterial catalysts intrinsically eliminates conventional catalyst-substrate heterointerfaces, efficiently preventing the potential peeling of active nanomaterial during long-term reaction. On the other hand, during the industry-level operating environment, the porous catalyst support is susceptible to diverse external compressive forces exerted by the electrode clips, water pressure, and adjacent components, leading to a diminution of its operational lifespan [R2.21]. To address this, it is necessary to further investigate the mechanical properties of our optimized FeCoNi dual-scale shell-lattice metamaterial, especially compared to commercial Ni foam (a commonly used support material). Therefore, we conducted *in situ* uniaxial compression experiments in SEM chamber to study the deformation behavior and mechanical response of FeCoNi dual-scale gyroid and random porous Ni foam.

In response to the reviewer's suggestions, we have extensively revised the first paragraph of the "*Practical applications of metamaterial catalysts*" section, with a particular focus on establishing a clear connection between mechanical tests and their implications for industrial-level catalytic applications. The revised version is shown below.

“Mechanical stability is a key indicator to evaluate the industrial applicability of electrocatalysts, however, existed research efforts appear disproportionately focused on performance optimization, with insufficient attention given to this essential aspect. Conventionally, the mechanical stability of electrocatalysts mainly involves the interface adhesion strength between support and nanomaterials and the intrinsic strength of porous catalyst support. On the one hand, the impact and drag forces induced by electrolyte convection and turbulence can cause catalyst detachment, which is a common problem especially for conventional powdery catalysts. On the other hand, during the industry-level operating environment, the porous catalyst support is susceptible to diverse external compressive forces exerted by the electrode clips, water pressure, and adjacent components, leading to a diminution of its operational lifespan. To address this, it is necessary to further investigate the mechanical

properties of our optimized FeCoNi dual-scale gyroid, especially compared to commercial Ni foam (a commonly used support material) under a consistent relative density of ~7.0 %.”

Comment 2.7

In Figure 6h, "interface stability" is listed as a characteristic for comparison between different catalysts, and the screw dislocation-driven metamaterial is shown to have the best performance. However, this is misleading because the screw dislocation-driven metamaterial is monolithic, inherently eliminating the catalyst-support interface. Therefore, "interface stability" is not applicable in this case. To avoid this confusion, please consider replacing "interface stability" with a more appropriate characteristic.

Response: Thank you for raising this possible confusion and your suggestion. We agree that the term "interface stability" may not be applicable to screw dislocation-driven metamaterials, as its monolithic structure inherently lacks conventional catalyst-support interfaces. To address this concern, we have revised the description in Fig. 6h and the corresponding text. The original “interface stability” has been replaced with “bonding strength” to reflect better the enhanced peeling resistance of monolithic metamaterial catalysts to mechanical stresses or degradation under operational conditions. In addition, to enhance clarity, the previous term "strength" has been revised to "structural strength" to better describe the mechanical loading capacity of the entire metamaterial in response to external compressive forces exerted by electrode clips, water pressure, and adjacent components.

According to the suggestion from the reviewer, we revised the related description throughout the manuscript accordingly, as shown below.

“Practical applications of metamaterial catalysts” section:

“Overall, our 3D-printed metamaterial catalysts, driven by screw dislocations, integrate multiple advantages, including design flexibility, structural strength, catalytic activity, and outstanding bonding strength, outperforming other 3D printing strategies and conventional randomly porous catalysts (Fig. 6h)”

“Summary” section:

“In summary, screw dislocations were employed to integrate the manufacturing of an efficient and stable metamaterial catalyst, featuring an interface-free surface nanostructures and ultrahigh dislocation storage.”

Fig. 6h Typical characteristics of metamaterial catalysts fabricated by various 3D printing strategies and conventional porous catalyst.

Comment 2.8

The manuscript highlights the advantage of a unified process to avoid heterointerface formation. However, the fabrication process involves multiple steps, including DLP printing of a hydrogel scaffold, immersion in a metal salt precursor, calcination to form metal oxides, and final reduction. Could the authors elaborate on which specific aspects of this multi-step process, and the resulting material characteristics, contribute to achieving an interface-free catalyst?

Response: To clarify the contribution of each step to achieving an interface-free catalyst, we provide a detailed schematic diagram of each step involved in the process and highlight their respective contributions and mechanisms, as shown in Fig. S3.

DLP 3D printing:

We added more details for the 3D printing hydrogel scaffold. First, we used DLP technology to 3D printing the organogel scaffolds and then soaked in deionized water for 6 h at room temperature to convert to hydrogel scaffolds. This step mainly aims to create highly designable 3D architectures as the support of the catalysts, which determines the macroscale topology morphology of final catalytic electrodes.

Metal salt precursor immersion:

The hydrogel scaffolds are immersed in a corresponding metal salt precursor solution, allowing Fe-Co-Ni ions to infuse into the architecture. This step is critical as it determines the final alloy composition of the catalysts. More importantly, the precursor concentration of metal salt solution in this step plays a crucial role in introducing screw dislocations during the subsequent sintering and

reduction process. First, we will provide some background on screw dislocation-driven nanomaterial growth. Since most catalytic nanostructures are designed to be anisotropic to maximize the exposure of electrochemically active sites, their formation process is fundamentally governed by crystal growth mechanisms [R2.22]. The challenge to grow anisotropic nanostructures is to break the symmetry in crystal growth. In many studies, using catalysts or templates is a widely employed approach to regulate crystal growth and achieve anisotropic nanostructures [R2.23-R2.25]. Screw dislocation mechanism is another strategy to synthesize anisotropic nanomaterials. According to the classical crystal growth theory, the supersaturation (σ) works as the driving force for crystal growth [R2.26].

$$\sigma = \ln \left(\frac{c}{c_0} \right)$$

where c and c_0 are the precursor concentration and equilibrium concentration of system. At low supersaturation conditions, screw dislocation defects can promote the crystal growth. The line of a screw dislocation creates step edges upon intersection with a crystal surface, which will propagate as self-perpetuating growth spirals. As supersaturation is increased, growth mode will transfer to layer-by-layer (LBL) growth and dendritic growth progressively [R2.27], as shown in Fig. S3b. Therefore, through intentionally exploiting low supersaturation conditions, many researchers achieved catalyst-free synthesis of a variety of anisotropic nanomaterials, including nanowires, nanoplates, and 3D hyperbranching nanostructures. Based on this theory, by utilizing low concentrations of Fe-Co-Ni metal salt solutions (2M), we have, for the first time, integrated this approach into the 3D printing process to *in situ* create helical nanoplates on the surface of the architecture. This step can be considered as the bridge between macroscale architecture fabrication and nanoscale nanomaterials synthesis.

Calcination and reduction:

The metal-salt-rich hydrogel scaffolds were calcinated in air to remove the polymer and convert to metal oxides. From the SEM image (Fig. S3a), we observed that Fe-Co-Ni oxides are clustered together in the form of nanoparticles. Finally, the metallic oxide nanoparticles were sintered, reduced, and transformed into a single solid-solution structure. This step can be considered as the final visualization process, where the metallic 3D framework is formed. At the same time, surface nanostructures are simultaneously grown on the architecture, directly determining the morphology of interface-free catalyst. The morphology feature of nanomaterial was considered to mainly depend on screw dislocation growth kinetics. At a low supersaturation, when growth velocities and directions of steps at the dislocation core (V_c) is equal to those at the outer edges (V_0), the newly generated steps near the dislocation core propagate at the same rate with earlier steps at the outer edge of the growth spiral [R2.28], inducing the growth of the nanoplates (Fig. S3c). Since the growth of the architecture and surface nanoplates is completed simultaneously during the reduction process, leading to the formation of an interface-free feature between the support and surface nanomaterials.

The related text was put into the supplementary information with a new *Supplementary Note 1*. We added a related sentence into the main manuscript, “*Screw dislocation-mediated integrated manufacturing of metamaterial catalysts*” section.

“The specific roles of each fabrication step in the development of the metamaterial catalyst are elaborated in Supplementary Note 1 and Fig. S3.”

Determining the architecture and composition

Introducing and controlling screw dislocation

Fig. S3 Mechanisms and contributions of each step in the fabrication process toward achieving interface-free metamaterial catalysts.

Comment 2.9

A higher concentration of NaNO_3 (0.5 M) was used for *in situ* Raman experiments. Could the authors please explain the reason behind using a higher concentration for the *in situ* Raman study? Furthermore, in the discussion of the *in situ* Raman results in Supplementary Note 2, it is stated that the antisymmetric bending vibration of adsorbed $^*\text{NH}_3$ at 1591 cm^{-1} disappeared when the potential was decreased to -0.5 V . However, based on Figure S28, this peak appears to become more prominent at -0.5 V . Could the authors please double-check the Raman spectra and revise the description accordingly?

Response: Thank you for your careful review and thoughtful suggestion. We chose 0.5 M NaNO_3 in *in situ* Raman Experiments because Raman signals from adsorbed intermediates (e.g., $^*\text{NO}_3$, and $^*\text{NO}$) are inherently weak due to low surface coverage under reaction conditions. A higher NaNO_3 concentration increases the adsorption flux of reactants, amplifying the signal of surface species for

clearer detection. However, this means that the *in situ* Raman test result cannot directly be comparable to that of nitrate reduction testing. To ensure comparability of results, the above mentioned *in situ* Raman tests were conducted in 0.1 M NaNO₃ and the laser power was increased to enhance Raman signals (Fig. S31a). Control tests confirmed that the lower NaNO₃ concentration does not alter the reaction mechanism compared to higher concentrations.

Fig. S31. (a) *in situ* Raman spectra of NO₃RR over of FeCoNi-S4650 at different applied potentials in 0.5 M Na₂SO₄ and 0.1 M NaNO₃.

We sincerely appreciate the reviewer's valuable feedback regarding the interpretation of the bending vibration of adsorbed *NH₃ in situ Raman results. In fact, what we wanted to express here is the symmetric bending vibrations of adsorbed *NH₃ at around 1591 cm⁻¹ appeared as the working potential shifted negatively to -0.5 V and the intensity of this peak further increased from -0.5 V to -0.9 V, which confirmed the formation of NH₃. We have revised the related description in *Supplementary Note 4* to ensure accuracy.

“The symmetric bending vibrations of adsorbed *NH₃ at around 1591 cm⁻¹ appeared as the working potential shifted negatively to -0.5 V and the intensity of this peak further increased from -0.5 V to -0.9 V, which confirmed the formation of NH₃.”

References

- [R2.1] Zhang, Rong, et al. "Efficient ammonia electrosynthesis and energy conversion through a Zn-nitrate battery by iron doping engineered nickel phosphide catalyst." *Advanced Energy Materials* 12.13 (2022): 2103872.
- [R2.2] Ye, Shenghua, et al. "Elucidating the activity, mechanism and application of selective electrosynthesis of ammonia from nitrate on cobalt phosphide." *Energy & Environmental Science* 15.2 (2022): 760-770.
- [R2.3] Fang, Jia-Yi, et al. "Ampere-level current density ammonia electrochemical synthesis using CuCo nanosheets simulating nitrite reductase bifunctional nature." *Nature Communications* 13.1 (2022): 7899.
- [R2.4] Yin, Di, et al. "Overcoming Energy-Scaling Barriers: Efficient Ammonia Electrosynthesis on High-Entropy Alloy Catalysts." *Advanced Materials* (2025): 2415739.

- [R2.5] Gao, Lei, et al. "Unconventional p-d hybridization interaction in PtGa ultrathin nanowires boosts oxygen reduction electrocatalysis." *Journal of the American Chemical Society* 141.45 (2019): 18083-18090.
- [R2.6] Lin, Gaoxin, et al. "Intrinsic electron localization of metastable MoS₂ boosts electrocatalytic nitrogen reduction to ammonia." *Advanced materials* 33.32 (2021): 2007509.
- [R2.7] Zhang, Shuo, et al. "Fe/Cu diatomic catalysts for electrochemical nitrate reduction to ammonia." *Nature Communications* 14.1 (2023): 3634.
- [R2.8] Wei, Min, et al. "High-entropy alloy nanocrystal assembled by nanosheets with d-d electron interaction for hydrogen evolution reaction." *Energy & Environmental Science* 16.9 (2023): 4009-4019.
- [R2.9] Gao, Qiang, et al. "Breaking adsorption-energy scaling limitations of electrocatalytic nitrate reduction on intermetallic CuPd nanocubes by machine-learned insights." *Nature communications* 13.1 (2022): 2338.
- [R2.10] Sun, Shanfu, et al. "Tailoring the d-band centers endows (Ni_xFe_{1-x})₂P nanosheets with efficient oxygen evolution catalysis." *Acs Catalysis* 10.16 (2020): 9086-9097.
- [R2.11] Wang, Siwen, Hemanth Somarajan Pillai, and Hongliang Xin. "Bayesian learning of chemisorption for bridging the complexity of electronic descriptors." *Nature communications* 11.1 (2020): 6132.
- [R2.12] Chen, Zhi Wen, et al. "Unusual Sabatier principle on high entropy alloy catalysts for hydrogen evolution reactions." *Nature Communications* 15.1 (2024): 359.
- [1] Li, Mingqiang, et al. "Harnessing dislocation motion using an electric field." *Nature Materials* 22.8 (2023): 958-963.
- [R2.13] Song, Yajing, et al. "Dynamic Homogenization of Internal Strain in Multi-Principal Element Alloy via High-Concentration Doping of Oxygen with Large Mobility." *Small Methods* 8.1 (2024): 2300871.
- [R2.14] Hao, Jican, et al. "Strain relaxation in metal alloy catalysts steers the product selectivity of electrocatalytic CO₂ reduction." *ACS nano* 16.2 (2022): 3251-3263.
- [R2.15] Gu, Jialun, et al. "Turing structuring with multiple nanotwins to engineer efficient and stable catalysts for hydrogen evolution reaction." *Nature Communications* 14.1 (2023): 5389.
- [R2.16] Wang, Yuting, et al. "Unveiling the activity origin of a copper-based electrocatalyst for selective nitrate reduction to ammonia." *Angewandte chemie international edition* 59.13 (2020): 5350-5354.
- [R2.17] Wang, Yuhang, et al. "Enhanced nitrate-to-ammonia activity on copper-nickel alloys via tuning of intermediate adsorption." *Journal of the American Chemical Society* 142.12 (2020): 5702-5708.
- [R2.18] Ye, Shenghua, et al. "Elucidating the activity, mechanism and application of selective electrosynthesis of ammonia from nitrate on cobalt phosphide." *Energy & Environmental Science* 15.2 (2022): 760-770.
- [R2.19] Liu, H. et al. Dual interfacial engineering of a Chevrel phase electrode material for stable hydrogen evolution at 2500 mA cm⁻². *Nature Communications* 13, 6382 (2022).
- [R2.20] Wang W, Yang Y, Zhao Y, et al. Multi-scale regulation in S, N co-incorporated carbon encapsulated Fe-doped Co₉S₈ achieving efficient water oxidation with low overpotential. *Nano Research*, 2022, 15(2): 872-880.
- [R2.21] B. Guo, J. Kang, T. Zeng, H. Qu, S. Yu, H. Deng, J. Bai, 3D Printing of Multiscale Ti₆Zr₄-Based Lattice Electrocatalysts for Robust Oxygen Evolution Reaction, *Advanced Science* 9(24) (2022) 2201751.

- [R2.22] Meng, Fei, et al. "Screw dislocation driven growth of nanomaterials." *Accounts of chemical research* 46.7 (2013): 1616-1626.
- [R2.23] Meng, Xiangyu, et al. "Hierarchical triphase diffusion photoelectrodes for photoelectrochemical gas/liquid flow conversion." *Nature Communications* 14.1 (2023): 2643.
- [R2.24] Persson, Ann I., et al. "Solid-phase diffusion mechanism for GaAs nanowire growth." *Nature materials* 3.10 (2004): 677-681.
- [R2.25] Xia, Younan, et al. "One-dimensional nanostructures: synthesis, characterization, and applications." *Advanced materials* 15.5 (2003): 353-389.
- [R2.26] Markov, Ivan Vesselinov. *Crystal growth for beginners: fundamentals of nucleation, crystal growth and epitaxy*. World scientific, 2016.
- [R2.27] Morin, Stephen A., et al. "Mechanism and kinetics of spontaneous nanotube growth driven by screw dislocations." *Science* 328.5977 (2010): 476-480.
- [R2.28] Chu, Yanhui, et al. "Morphological control and kinetics in three dimensions for hierarchical nanostructures growth by screw dislocations." *Acta Materialia* 162 (2019): 284-291.

Reviewer #3

General comment:

This paper describes a method for creating advanced catalysts using 3D printing technology. It focuses on screw dislocations to enhance both the manufacturing process and the catalytic performance of the FeCoNi shell-lattice metamaterials. By inducing strain effects, these dislocations improve nitrate-to-ammonia conversion efficiency. The concept of utilizing screw dislocations in 3D printing to enhance catalytic performance is innovative and can bridge the gap between nanoscale material design and macroscale manufacturing. Here are some comments and questions.

Response: We sincerely thank the reviewer for the positive feedback and comments on our manuscript, and especially for recognizing our new concept of utilizing screw dislocations in 3D printing to develop efficient and stable metamaterial catalysts. Screw dislocation was utilized as a bridge to connect the macroscale 3D manufacturing and low-dimensional nanomaterials synthesis. Below, we address each specific comment in detail, aiming to improve the clarity and thoroughness of the manuscript. We believe these revisions will enhance the overall quality of the manuscript and align it with the standards required for publication.

Comment 3.1

Could you provide more details on how the screw dislocations are introduced and controlled during the 3D printing process?

Response: To answer this question, we will give more details about the introduction and control of screw dislocation growth mode during 3D printing process (since part of this question overlaps with *comment 2.8* from Reviewer #2, some text of the response may appear repetitive).

Introducing screw dislocation growth behavior:

When hydrogel scaffolds are immersed in a corresponding metal salt precursor solution, Fe-Co-Ni ions infuse into the architecture (Fig. S3a). The precursor concentration of the metal salt solution plays a crucial role in facilitating the introduction of screw dislocations during the subsequent sintering and reduction processes. We will provide some background on screw dislocation-driven nanomaterial growth. Since most catalytic nanostructures are designed to be anisotropic to maximize the exposure of electrochemically active sites, their formation process is fundamentally governed by the mechanisms of crystal growth [R3.1]. The challenge to grow anisotropic nanostructures is to break the symmetry in crystal growth. In many studies, the use of catalysts or templates is a widely employed approach to regulate crystal growth and achieve anisotropic nanostructures [R3.2-R3.3]. Screw dislocation mechanism is another strategy to synthesize anisotropic nanomaterials. According to the classical crystal growth theory, the supersaturation (σ) works as the driving force for crystal growth [R3.4].

$$\sigma = \ln \left(\frac{c}{c_0} \right)$$

where c and c_0 are the precursor concentration and equilibrium concentration of system. At low c

conditions, screw dislocation defects can promote the crystal growth. The line of a screw dislocation creates step edges upon intersection with a crystal surface, which will propagate as self-perpetuating growth spirals. As supersaturation is increased, growth mode will transfer to layer-by-layer (LBL) growth and dendritic growth progressively [R3.5], as shown in Fig. S3b. Therefore, through intentionally exploiting low supersaturation conditions, many researchers achieved catalyst-free synthesis of a variety of anisotropic nanomaterials, including nanowires, nanoplates, and 3D hyperbranching nanostructures. Based on this theory, by utilizing low concentrations of Fe-Co-Ni metal salt solutions (2M), we have, for the first time, integrated this approach into the 3D printing process to *in situ* create helical nanoplates on the surface of the architecture.

Controlling screw dislocation growth behavior:

Due to the strong correlation between screw dislocations and the formation of surface nanoplates, we can analyze the surface morphology to gain insights into the growth mechanism of the sample. Based on the discussion above, we conducted a comparative experiment by increasing the concentration of the Fe-Co-Ni salt solution by 5 times. In this case, we observed that most grains exhibited a smooth surface morphology with very few nanosteps (Fig. S4). This finding indicates a transition in the growth mode from a screw dislocation-driven mechanism to a layer-by-layer growth mechanism. Additionally, we observed that the sintering duration during the hydrogen reduction process significantly influences the growth mode of the sample. Both excessively short and overly long growth times are detrimental to maintaining the screw dislocation growth mode. Therefore, in the manuscript, we selected a concentration of 2 M and a sintering duration of 8 hours as the optimal parameters for the precursor solution and sintering process, respectively. Since the growth of the architecture and surface nanoplates is completed simultaneously and driven by screw dislocation, leading to the screw dislocation-rich surface nanomaterials in architecture. We need to note that a correction is required in the previous description of the metal salt solution concentration. The original statement, “0.1M and 0.5M metal salt solution of nickel nitrate, iron nitrate, and cobalt nitrate for FeCoNi alloy were prepared with deionized water”, should be revised to “2 M and 10 M metal salt solution of nickel nitrate, iron nitrate, and cobalt nitrate for FeCoNi alloy were prepared with deionized water.”.

Since part of this question overlaps with comment 2.8 from Reviewer #2, we have addressed it by adding a new figure and *Supplementary Note 1* to provide a detailed description of the introduction and control process of screw dislocation during the fabrication steps.

Determining the architecture and composition

Introducing and controlling screw dislocation

Fig. S3 Mechanisms and contributions of each step in the fabrication process toward achieving interface-free metamaterial catalysts.

Fig. S4 Synthesis parameters optimization of metal salt precursor concentration and reduction duration.

Comment 3.2

How does the dislocation density compare to that of conventionally manufactured catalysts?

Response: In fact, it is somewhat challenging to precisely compare the dislocation density of metamaterial catalysts with those reported in previous literatures. First, most previous reports on dislocation-rich catalysts primarily focus on observing the presence of dislocations and their associated strain effects in various electrocatalytic reactions (*Nature Materials* 20.7 (2021): 1000-1006; *Advanced Materials* 34.2 (2022): 2106973; *Advanced Materials* 32.48 (2020): 2006034). However, very few studies have provided detailed quantification of dislocation density, making direct comparisons challenging. Second, the statistical methods used to quantify dislocation density vary significantly across different studies, making it difficult to establish a consistent basis for comparison. For example, several studies, particularly those focusing on conventional nanomaterials, have employed geometric phase analysis (GPA) of HRTEM images to quantitatively assess dislocation density. In this case, dislocation density is defined as the number of dislocations intersecting a unit area, which was calculated by using Equation [R3.6-R3.7]:

$$\rho = \frac{n}{S}$$

where n is the number of dislocations pairs in GPA image, and S is the cross-sectional area of HRTEM images. As a localized characterization technique, the width of HRTEM images is limited to just a few nanometers, posing significant challenges in accurately capturing the true dislocation density of samples. Consequently, statistical results derived from this method tend to be relatively high. In contrast, Williamson-Hall and EBSD techniques are mesoscale tools capable of sampling dislocation densities across millions of grains [R3.8], providing a more comprehensive and representative overview of the material's characteristics.

Nevertheless, according to the suggestion from the reviewer, we used two-scale methods to compare the dislocation density at mesoscale and nanoscale. In addition to the Williamson-Hall and EBSD methods utilized in the manuscript, here, we calculated the dislocation density according to GPA image of HRTEM in Fig. 2f. To ensure a fair comparison, we classified the measurement scale into two categories and compared our results with previously reported dislocation density of conventionally manufactured catalysts, as shown in Table S2. It can be observed that, regardless of the statistical standard employed, our sample both exhibited comparable or even higher dislocation density. More importantly, we need to emphasize that the dislocation density is not our biggest highlight. This study aims to introduce a screw dislocation-driven 3D printing strategy for fabricating monolithic metamaterial catalysts with high design freedom and interface-free nanomaterials. Our approach represents a significant advancement in bridging the gap between nanoscale catalyst material design and macroscale 3D-architected electrode manufacturing, showcasing the potential for industrial applications.

We therefore revised the related text in “*dislocation analysis in shell-lattice metamaterials*” section to display the comparison of dislocation density with previous literatures.

“Compared to previous catalysts fabricated by conventional methods, our 3D printed metamaterial catalysts exhibited comparable or even higher dislocation density (Table S2)”

Table S2. Dislocation density comparison using different measurement methods

Estimation method	Material	Catalytic reaction	Dislocation density (m^{-2})	References
Williamson-Hall or EBSD (macro or mesoscale)	FeCoNi	Electrochemical nitrate reduction reaction	2.23×10^{14} (WH) 1.35×10^{14} (EBSD)	This work
	BaTiO ₃	Water splitting	1.5×10^{14}	R3.9
	Pure Cu	CO ₂ reduction reaction	2.23×10^{14}	R3.10
	NiCrFeMo	CO ₂ methanation	1.32×10^{14}	R3.11
GPA analysis of HRTEM (nanoscale)	FeCoNi	Electrochemical nitrate reduction reaction	2.2×10^{17}	This work
	Mo ₂ C	Hydrogen evolution reaction	1.0×10^{17}	R3.12

	V ₂ O ₅	Hydrazine oxidation	1.5×10^{15}	R3.13
	PtNi/NF	Hydrogen evolution reaction	9.9×10^{17}	R3.7

Comment 3.3

The authors mention that "78% of the dislocations were screw-type." Is it possible to provide more insight into how this percentage was determined from the STEM analysis?

Response: According to some previous studies (*Nature Communications*, 12(1), 5474; *Nature Communications*, 2022, 13(1): 6449.; *Acta Materialia*, 2016, 110: 352-363), we employed similar methods to determine the type and percentage of dislocation via BF-STEM. First, we will introduce some relevant background and mechanism. Diffraction contrast TEM is a powerful technique for imaging dislocations in crystals, leveraging additional electron diffraction caused by the bending of atomic planes near the dislocation core. If an image is reconstructed using specific reciprocal space diffraction spots (g) selected by a physical aperture, these additional diffracted electrons create a visible contrast around the dislocation. However, diffraction spots (g) that are oriented perpendicular to the Burgers vector do not produce any dislocation contrast, which is known as the "invisibility criterion" [R3.14]. Therefore, the direction of a Burgers vector (b) can be identified by taking the cross product of two noncollinear g vectors [R3.15]. Subsequently, by analyzing the directional relationship between the dislocation line observed in the TEM image and the Burgers vector b , the type of dislocations can be determined. Specifically, if the dislocation line is perpendicular to b , it is identified as an edge dislocation, whereas if the dislocation line is parallel to b , it is classified as a screw dislocation [R3.16]. By analyzing the dislocations distribution in one TEM image, the relative proportion of edge and screw dislocations can be quantified.

Based on the analysis above, the TEM sample was first tilted close to the [110] zone axis. The image with the $(0\bar{2}2)$ and $(\bar{1}11)$ diffraction spot (Fig. 4d) shows strong dislocation contrast, while the dislocation meets the invisibility criterion under the $(1\bar{1}1)$ family of spot. Then, the Burgers vector was determined to be [110] direction. Subsequently, to enhance the image contrast, we enter corresponding STEM mode to directly observe the dislocation line orientation at two-beam condition. All visible dislocations are identified. The identified line direction is compared with the Burgers vector direction. If the difference is <10 degrees, then the dislocation character is determined. However, most dislocation in natural crystal is curved even entangled, pure edge and screw dislocations can be considered as the idealized extremes [R3.17]. Normally, a curved dislocation line contains both edge and screw components. In this case, according to widely used dislocation analysis method (*Nature Communications*, 12(1), 5474; *Science* 370.6512 (2020): 95-101; *Acta Materialia* 144 (2018): 107-115), if two characteristics are possible, the one with the closest match to the dislocation line is chosen. If the dislocation line does not meet these criteria, it is not considered. Therefore, the type of dislocation lines in a typical TEM image can be determined and then their percentage can be calculated.

Combining the *comment 3.5*, we revised the "dislocation analysis in shell-lattice metamaterials" section to clarify and add more details, as shown below.

"Diffraction contrast TEM is a powerful technique for imaging dislocations in crystals, leveraging additional electron diffraction caused by the bending of atomic planes near the dislocation core."

“The identified line direction is compared with the Burgers vector direction. If the angular difference between the Burgers vector and the dislocation line is less than 10 degrees, then the dislocation character is determined. If two characters are possible, the one with the closest match to the dislocation line is chosen. Therefore, multiple dislocation arrays in the same plane along the growth axis was confirmed to be close to pure screw dislocations (Fig. 2c). A few dislocations are observed near the edge character (Fig. 2c). Statistical analysis from corresponding STEM image further revealed that 78% of the dislocation were screw-type, while 22% were edge-type, indicating that the growth of nanosteps was governed by screw dislocation.”

Comment 3.4

The dislocation density was estimated via XRD diffraction peak broadening using the Williamson-Hall method and also measured through EBSD. A clarification on how the two methods compare or validate each other could strengthen the manuscript.

Response: We appreciate the reviewer’s insightful comment regarding comparing the dislocation density obtained from the Williamson-Hall method (WH) and EBSD. First, we will give more measurement details of two methods.

For Williamson-Hall method [R3.18]:

Instrument broadening of XRD was calibrated using a strain-free silicon powder as the reference.

$$\delta \cos\theta = \frac{\lambda}{D} + 2\varepsilon \sin\theta$$

where δ is the physical broadening of full width at half maximum (FWHM) of the diffraction peak, θ is the diffraction angle, λ is the wavelength of radiation, D is the grain size, and ε is the internal strain. The linear fit may be used to get the ε and D . Grain size has a negligible effect on peak broadening. (the wavelength $\lambda_{K\alpha 1}=0.154\text{nm}$ and $D\sim 2.8\mu\text{m}$). Hence, the term of λ/D can be close to zero. For linear fitting, three planes of (111), (200), and (220) were used. The dislocation density ρ was calculated as below:

$$\rho = k \frac{\varepsilon^2}{b^2}$$

where $k=16.1$ for FeCoNi [R3.19], and the Burgers vector b is 0.2525nm [R3.20].

For EBSD method [R3.21]:

Geometrically necessary dislocation (GND) was obtained from the Kernel Average Misorientation (KAM) values of EBSD using the following equation. KAM maps were generated by calculating the misorientation between the central pixel of the kernel and its first nearest neighbors, as shown in Fig. 2e. A threshold misorientation angle of 5° was applied to exclude the influence of high-angle grain boundaries [R3.22].

$$\rho_{GND} = \frac{\alpha_{gb} \theta_{KAM}}{bx}$$

Where θ_{KAM} is the KAM value, b is burgers vector, x is unit length and α_{gb} is a constant value.

First, we compared the implications of the results calculated by these two methods. As an X-ray

diffraction (XRD) technique, the Williamson-Hall (WH) method was well-accepted to determine the total dislocation density by analyzing XRD peak broadening, which captures contributions from both microstrain and crystallite size. According to the report by Ashby [R3.23], the dislocation density of WH method is the sum of GNDs and statistically stored dislocation (SSDs), providing a more comprehensive measure of the total dislocation density. It should be noted that EBSD detects the geometrically necessary portion of the dislocation population (GNDs) and is generally blind to the statistically stored portion of the dislocation population (SSDs). It indicates that the GND density measured by EBSD is typically lower than the dislocation density estimated by the Williamson-Hall method in a typical sample. Thus, the EBSD results can serve as an indirect validation of the accuracy of the WH results. For this reason, in the subsequent comparative experiments involving different Gaussian curvatures, we use the WH results for comparison, as they provide a complete and representative measure of the total dislocation density.

Then, regarding the analysis scale of the two methods, XRD provides a volume-averaged value at the macroscale, offering a bulk measurement of dislocation density across the entire sample. In contrast, EBSD serves as a mesoscale tool but with high spatial resolution, enabling the visualization of dislocation-induced strain distribution and localized variations in lattice misorientation. This complementary combination of macroscale (WH) and mesoscale (EBSD) analyses allows for a more comprehensive understanding of dislocation behavior.

In fact, in previous studies, many papers have also employed two methods, the Williamson-Hall and EBSD, to compare and validate dislocation density measurements [R3.24-R3.25]. The mutual complementation not only enhances the reliability of dislocation density quantification but also provides deeper insights into the material's dislocation behavior from multiple scales and visualization angles.

We slightly revised the Fig. 2d to clarify the meaning of WH and EBSD measurement and revised related sentence, as shown below.

“Based on XRD diffraction peak broadening effect, we employed the macroscale Williamson-Hall method to estimate total dislocation density of $2.23 \times 10^{14} \text{ m}^{-2}$,”

Fig. 2d Statistical result revealing the dislocation density and the percentage of different dislocation types.

Comment 3.5

It would be interesting to know the proportion of mixed dislocations relative to screw and edge dislocations and whether these contribute differently to the catalytic activity or structural properties.

Response: Thanks for the interesting question. First, it is essential to clarify the relationship between mixed dislocations and screw/edge dislocations. In theory, edge dislocations and screw dislocations are considered the two fundamental types of dislocations. However, in natural crystals, most dislocations are curved, meaning that pure edge and screw dislocations only can be viewed as idealized extremes. In reality, the majority of dislocations in crystals are mixed, displaying characteristics of both edge and screw components. Consequently, if we strictly define mixed dislocation, pure edge and screw dislocations will hardly be observed in practice. Therefore, mixed dislocations should not be discussed separately, as they inherently combine both edge and screw dislocation components, as illustrated in the schematic diagram (Fig. R3). In such cases, studies involving dislocation analysis typically simplify the discussion by categorizing dislocations as either screw or edge, based on which type of dislocation component is predominant to characterize the dislocation line. Similarly, in the current work, the morphology of nearly all dislocations in Fig. 2 is curved and even entangled. Specifically, if the angular difference between the Burgers vector and the dislocation line is less than 10 degrees, then the dislocation character is determined. If two characters are possible, the one with the closest match to the dislocation line is chosen. This method has also been widely recognized and employed in many studies (*Nature Communications*, 12(1), 5474; *Science* 370.6512 (2020): 95-101; *Acta Materialia* 144 (2018): 107-115).

To avoid potential misunderstanding, we have revised the manuscript for clarification in “Dislocation analysis in shell-lattice metamaterials” section.

“The identified line direction is compared with the Burgers vector direction. If the angular difference between the Burgers vector and the dislocation line is less than 10 degrees, then the dislocation character is determined. If two characters are possible, the one with the closest match to the dislocation line is chosen.”

Fig. R3 The schematic diagram of mixed dislocations with screw and edge components.

Then, we address the comments regarding the influence of screw and edge dislocations on catalytic activity, respectively. Screw dislocations induce spiral growth patterns (the surface step defects), which enhance the electrochemically active area. Advancing the progress of NO_3^- reduction with a lower thermodynamic barrier of 0.51 eV as compared to that of none stepped FeCoNi-LBL (0.89 eV) (Fig. 5e). Although computational results indicate that screw dislocations dominate in this catalyst, as noted by the reviewer, edge dislocations also play a non-negligible role in modulating catalytic activity. Therefore, we employed density functional theory (DFT) to investigate the structure of edge dislocations and their impact on catalysis (Fig. S38). As a result, the potential-determining step (PDS) on the FeCoNi slab with edge dislocations is the $^*\text{NH}_2\text{OH}$ to $^*\text{NH}_2$ step, with a significantly lower thermodynamic barrier of 0.65 eV. It means that these exposed sites created by edge dislocations reduce the energy barrier of NO_3RR as compared to FeCoNi-LBL. Moreover, severe strain effects induced by screw and edge dislocation enhance intrinsic catalytic activity by promoting NO_3^- adsorption and lowering the energy barrier of NO_3^- -to- NH_3 conversion. In that way, those two types of dislocations contribute similarly to the catalytic activity.

We have added the related description and in “*Electrocatalysis performance and mechanism*” section, and figure was put into supplementary information.

“Although screw dislocations remain the predominant type across substrates with varying curvatures, the role of edge dislocations should also be studied in modulating catalytic activity. According to the density functional theory calculation of edge dislocations (Fig. S38), the potential-determining step on the FeCoNi slab with edge dislocations is the $^*\text{NH}_2\text{OH}$ to $^*\text{NH}_2$ step, with a significantly lower thermodynamic barrier of 0.65 eV. These exposed sites created by edge dislocations reduce the energy barrier of NO_3RR as compared to FeCoNi-LBL.”

Fig. S38 (a) The adsorption models of various intermediates generated during NO₃RR pathways on the surface of FeCoNi slab edge dislocations. (b) Free-energy diagram for NO₃RR on FeCoNi slab with edge dislocations.

Comment 3.6

It would be valuable to explain how the lattice strain directly impacts the catalytic activity. For instance, how does a 6.3% lattice tension (as seen in the highest curvature region) influence the material's catalytic efficiency or stability?

Response: Thank you for your insightful question regarding the direct impact of lattice strain on catalytic activity. Below, we elaborate on how the observed 6.3% lattice tension in high-curvature regions enhances catalytic efficiency:

To elucidate the mechanism behind the enhanced NO₃RR activity, we performed theoretical calculations on the monolithic metamaterial catalysts, characterized by lattice strain induced by dense dislocation network. First, the stepped FeCoNi under 6% tensile strain exhibits the most favorable NO₃⁻ adsorption with Gibbs free energy changes ($\Delta G^*_{\text{NO}_3}$) ranging from -0.01 to -1.26 eV, as compared

to that of stepped FeCoNi (from 0.54 to -1.0 eV) and stepped FeCoNi under 3% tensile strain (from 0.36 to -0.95 eV) (Fig. 5b). This strong adsorption of NO_3^- lowers the energy barrier of the initial discharge step ($^*\text{NO}_3^- \rightarrow ^*\text{NO}_3 + \text{e}^-$). This result is also evidenced by the largest charge transfer in FeCoNi with 6% strain, which is induced by an upward shift of 3d-band center (Fig. 5d). Furthermore, 6% lattice tension created by screw dislocation-driven growth also exhibits the lowest thermodynamic barrier (0.3 eV) in the following deoxidation/hydrogenation steps in NO_3RR (Fig. 5e). Therefore, introducing screw dislocation can dramatically enhance the catalytic activity for NO_3RR , promoting NH_3 synthesis.

Comment 3.7

The comparative analysis between dual-scale shell lattice and reference metamaterials (simple cubic and single-order shell lattice) is valuable for demonstrating the effects of curvature. However, it would be helpful to expand on how these materials' differing mean absolute Gaussian curvatures (MAGC) specifically influence their catalytic properties. Are the increased dislocation densities in the curved metamaterials contributing to higher catalytic efficiency, and if so, how?

Response: Besides the strain energy created by screw dislocation growth, the introduction of substrate curvature will generate additional elastic energy, with its magnitude increasing remarkably as the curvature. The elevated energy can be released by disrupting the crystal's symmetry by multiplying dislocations at the curved surface. Consequently, increased mean absolute Gaussian curvatures result in a substantial rise in dislocation density. As shown in Fig. S14, only a few dislocation lines, primarily composed of screw dislocations, are observed in regions with small curvature. In contrast, a high-density dislocation network was observed to entangle and form an intricately interwoven network in areas with large curvature.

Defect engineering has been regarded as an effective strategy for boosting the intrinsic catalytic activity in various electrochemical reactions through tuning electronic structure of active sites and thus optimizing intermediates chemisorption on catalysts surface [R3.26-R3.27]. As a typical line defect, the presence of dense dislocation network can generate serious and inhomogeneous lattice strain fields. We observed up to 6.3% of lattice strains in nanomaterials at regions of high curvature in dual-scale shell-lattice metamaterials. Many studies in the existing literature have demonstrated that high-energy surfaces with dislocation-induced strain effects can modulate the electronic structure and enhance catalytic activity across various catalytic reactions, including hydrogen evolution reaction (HER) [R3.28-R3.29] and formate oxidation reactions (FOR) [R3.30]. Furthermore, in nitrate-to-ammonia reduction reaction (NO_3RR) experiment, we observed that increased dislocation density can promote NO_3^- -to- NH_3 conversion at high efficiency and lower energy consumption, leading to outstanding current density, Faraday efficiency, and NH_3 yield rate.

To further elucidate the underlying mechanism by which dislocations enhance catalytic activity, we conducted theoretical calculations. With the introduction of lattice strain, we found that FeCoNi with 6% strain from dislocation shows the highest charge transfer value (0.83 e^-), offering strong evidence of NO_3^- adsorption (Fig. 5b). D-band center model suggests that the lattice strain in FeCoNi-LBL slab induce an upward shifted of 3d-band center (Fig. 5d). Lattice strain in stepped FeCoNi slab retains the same PDS ($^*\text{NH}_2\text{OH}$ to $^*\text{NH}_2$) while further advancing the progress of NO_3^- reduction with a lower energy barrier of 0.3 eV (Fig. 5e). The lower thermodynamic barrier of NO_3RR could markedly assist

the following deoxidation/hydrogenation steps in NO₃RR. Therefore, increased dislocation density in curved metamaterials can dramatically enhance the catalytic activity for NO₃RR, promoting NH₃ synthesis.

Comment 3.8

Given the challenges involved in calculations, how did the authors calculate the energy of the charged NO₃⁻ ion?

Response: We appreciate the reviewer's question regarding the calculation of the energy of the charged NO₃⁻ species.

The Gibbs free energy of NO₃ adsorption on the surface (*NO₃) (Eq.(S6)), following the equation (1).

To avoid obtaining the energy of charged NO₃⁻ species directly, the neutral HNO₃ gas phase was chosen to be a reference as below equation (2-3) [R3.31][R3.32]:

Hence, the equation (1) can be rewritten as below (4):

The Gibbs free energy of NO₃ ($\Delta G(*\text{NO}_3)$) can be calculated as (5):

$$\Delta G(*\text{NO}_3) = G(*\text{NO}_3) - G(\text{HNO}_3) + \frac{1}{2}G(\text{H}_2) - G(*) + \Delta G_{\text{correct}} \quad (5)$$

in which $G(*\text{NO}_3)$, $G(\text{HNO}_3)$ and $G(*)$ are the Gibbs free energy of adsorbed nitrate, vaporization of HNO₃ (l) and catalysts, respectively. $\Delta G_{\text{correct}}$ is the correction of adsorption energy [R3.33].

$$\Delta G_{\text{correct}} = \Delta G_{\text{S1}} + \Delta G_{\text{S2}} \quad (6)$$

ΔG_{S1} is the Gibbs free energy of formation of HNO₃(l) from NO₃⁻ (l) (equation 2: $\Delta G_{\text{S1}} = 0.317$ eV). ΔG_{S2} is the Gibbs free energy of vaporization of HNO₃(l), which was calculated from the Gibbs free energy difference between the HNO₃(l) and HNO₃(g) (equation 3 : $\Delta G_{\text{S2}} = 0.074$ eV).

The updated results, which include solvation effects, are presented in *Computational methods* of the revised manuscript. We believe these improvements significantly enhance the reliability and relevance of our findings.

Comment 3.9

The elementary reaction steps of NO₃RR described in the Computational Methods section contradicts the Free-energy diagram presented in Figure 5e, which requires careful modification for consistency.

Response: We sincerely thank the reviewer for pointing out this oversight. To address this issue, we have made the following modifications:

The NO₃⁻ reduction process can be described as following steps (1-9):

Where the * represents the active site.

For each step, the reaction free energies can be obtained after gas correction as follows (10):

$$\Delta G = \Delta E + \Delta E_{\text{ZPE}} - T\Delta S \quad (10)$$

where ΔE is the total energy difference between the reactant and the product. ΔE_{ZPE} and ΔS are, respectively, the change in the zero-point energy and entropy, which are obtained from the vibrational frequency calculations. T is the temperature (T= 298.15 K).

Please find the corrected version in *Computational methods* of the revised manuscript.

Comment 3.10

Could the authors expand on how dislocations and strain specifically affect other intermediate steps, such as *NO and *NH₂?

Response: We appreciate the reviewer for raising this critical and challenging question regarding the specific roles of dislocations and strain in modulating intermediate steps during the reaction. To elucidate the effect of dislocations and strain on other intermediates (*NO and *NH₂), the Gibbs free energies of each intermediate analyzed in deoxidation and hydrogenation steps (Fig. 5e and Fig.S36-37). FeCoNi-LBL and stepped FeCoNi exhibit various uphill free energy increments in the electron-proton transfer. The *NO hydrogenation step (*NO + H₂O + e⁻ → *NOH + OH⁻) was the potential-determining step (PDS), involving a ΔG of 0.89 eV over FeCoNi-LBL. In contrast, the PDS on the stepped FeCoNi slab is the *NH₂OH to *NH₂ step, with a significantly lower thermodynamic barrier of 0.51 eV. This PDS change is due to the exposed step sites created by screw dislocation-driven growth, which reduce the free energy of *NO to *NOH from 0.89 eV to 0.11 eV. Moreover, lattice strain in stepped FeCoNi slab retains the same PDS (*NH₂OH to *NH₂) while further advancing the progress of NO₃⁻ reduction with a lower energy barrier of 0.3 eV. The lower thermodynamic barrier of NO₃RR could markedly assist the following deoxidation/hydrogenation steps in NO₃RR. Therefore, introducing screw dislocation can dramatically enhance the catalytic activity for NO₃⁻-to-NH₃ conversion.

Although those detailed theoretical calculations predict that dislocations and strain enhance the generation of other intermediates (*NO and *NH₂) in multi-step NO₃RR, this conclusion hasn't been verified in the experiments part. As suggested by the reviewer, we supplemented the related observations by *in situ* Raman spectroscopy. The Raman scattering peaks observed in NO₃RR on FeCoNi-LBL and stepped FeCoNi with 6% strain. With the decreasing of potential from open circuit potential (OCP) to -0.9 V, several peaks came out in sequences in stepped FeCoNi with 6% strain catalysts (Fig. S31a). At open circuit potential (OCP), the initial characteristic peaks at around 990 cm⁻¹ and 1049 cm⁻¹ were attributed to SO₄²⁻ and NO₃⁻ stretching, respectively. With the increase of the applied voltage, the *NO₃⁻ peak appeared around 1030 cm⁻¹, indicating the adsorption of NO₃⁻ on the surface of stepped FeCoNi with 6% strain. Moreover, the peaks of NO stretching vibration emerged near 998 cm⁻¹, and symmetric bending vibrations of HNH were visible near 1315 and 1340 cm⁻¹ at -0.5 V. The symmetric bending vibrations of adsorbed *NH₃ at around 1591 cm⁻¹ appeared as the working potential shifted negatively to -0.5 V, which confirmed the formation of NH₃ [R3.34]. The

Raman spectra on FeCoNi-LBL catalyst indicates that the peak related to *NO, *NH₂ and *NH₃ appeared at higher overpotential (−0.6 V) due to its poor ability for deep *NO₃[−] hydrodeoxygenation properties. Moreover, the band's intensities of all the detected intermediates formed on FeCoNi-LBL were very weak, meaning that a low amount of intermediates (*NO and *NH₂) and product (*NH₃) were generated (Fig. S31b). These experimental and theoretical results confirm that stepped FeCoNi with 6% strain could lower the energy barrier of NO₃RR and activate NO and NH₂ molecules, leading to a high NH₃ yield and selectivity. The revised sentences and figures can be shown below.

“Supplementary Note 4” section

“The Raman spectra on FeCoNi-LBL catalyst indicates that the peak related to *NO, *NH₂ and *NH₃ appeared at higher overpotential (−0.6 V) due to its poor ability for deep *NO₃[−] hydrodeoxygenation properties. Moreover, the band's intensities of all the detected intermediates formed on FeCoNi-LBL were very weak, meaning that a low amount of intermediates (*NO and *NH₂) and product (*NH₃) were generated (Fig. S31).”

Fig. S31. *in situ* Raman spectra of NO₃RR over of FeCoNi-S4650 (a) and FeCoNi-LBL (b) at different applied potentials in 0.5 M Na₂SO₄ and 0.1 M NaNO₃.

Comment 3.11

Did the author consider solvation effect in their calculations? It is quite important factor to be considered in these electrocatalysis.

Response: Thanks for raising this important point regarding considering solvation effects in our calculations. We agree that solvation effects are indeed crucial in electrocatalytic systems, as they can significantly influence the energetics and reaction pathways. In the current study, solvation effects were excluded from calculations to focus on characterizing surface step defects and heterogeneous lattice strain induced by dense dislocation networks. However, we acknowledge that this is an important aspect that should be addressed to further improve the accuracy of our results.

In response to the reviewer's comment, we have now incorporated solvation effects using *ab initio* molecular dynamics (AIMD) simulations in our revised calculations. Under the neutral electrolysis condition, these nitrate readily integrate into the hydrogen bond network formed by interfacial water, theoretically facilitating the reduction of nitrate via the Eley-Rideal (ER) mechanism directly interacting with the H₂O (*M + H₂O + e[−] = *MH + OH[−] mechanism rather than the Langmuir-Hinshelwood (LH) mechanisms via *H (*M + *H = *MH) mechanism involving *H

participation[R3.35-R3.37]. In that way, we investigated the hydrogen transfer process for water molecules at the electrolyte-catalyst interface for FeCoNi-LBL and stepped FeCoNi with 6% strain using the Polarized Continuum Model (PCM). To perform simulation under 0.1 M nitrate concentrations, a model with 1 nitrate and 315 water molecules was constructed. Fig. R4 presents the equilibrium water density profiles normal to the catalyst surface obtained from 10 ps AIMD simulations. The planes consisting of water molecules closest to the stepped FeCoNi with 6% strain model is 4.87 Å (from the nearest water to the bottom of the model), this is 1.73 Å nearer than the FeCoNi-LBL model. This difference is attributed to the surface step defects in the stepped FeCoNi with 6% strain model. While the density of water in the surface water region is similar in the stepped FeCoNi with 6% strain model compared to FeCoNi-LBL. The comparable water concentration surrounding nitrate ions leads to a similar hydrogen bond network between the nitrate ions and water molecules, indicating analogous solvation effects. To further confirm this, the average number of hydrogen bonds between nitrate and water molecules is similar in both FeCoNi-LBL and stepped FeCoNi with 6% strain slab over the 10 ps duration. These results indicate that the effects of solvent environment on FeCoNi-LBL and the stepped FeCoNi with 6% strain, induced by screw dislocation, are similar. Additionally, the enhanced catalytic performance is heavily relied on the activation of nitrate and other intermediate species on the catalysts. Thus, the work is intentionally focused on enhancing intrinsic catalytic activity by the dislocation multiplication, as our primary aim is to propose a screw dislocation-mediated dual-scale shell-lattice metamaterials. We have noted this valuable suggestion and plan to explore solvation effect in depth as part of our upcoming work.

Fig. R4. (a) Distribution of water molecules in the X - Y plane along the z axis electrode distance from the catalyst; (b) representative snapshots after 10 ps simulation for interfacial water molecules using AIMD method; (c) the number of collisions between oxygen atoms in nitrate and water molecules.

References

- [R3.1] Meng, Fei, et al. "Screw dislocation driven growth of nanomaterials." *Accounts of chemical research* 46.7 (2013): 1616-1626.
- [R3.2] Meng, Xiangyu, et al. "Hierarchical triphase diffusion photoelectrodes for photoelectrochemical gas/liquid flow conversion." *Nature Communications* 14.1 (2023): 2643.
- [R3.3] Persson, Ann I., et al. "Solid-phase diffusion mechanism for GaAs nanowire growth." *Nature materials* 3.10 (2004): 677-681.
- [R3.4] Xia, Younan, et al. "One-dimensional nanostructures: synthesis, characterization, and applications." *Advanced materials* 15.5 (2003): 353-389.
- [R3.5] Markov, Ivan Vesselinov. *Crystal growth for beginners: fundamentals of nucleation, crystal*

growth and epitaxy. World scientific, 2016.

- [R3.6] Stráský, J., et al. "Microstructure and lattice defects in ultrafine grained biomedical α + β and metastable β Ti alloys." Titanium in medical and dental applications. Woodhead Publishing, 2018. 455-475.
- [R3.7] Zhou M, Cheng C, Dong C, et al. Dislocation Network-Boosted PtNi Nanocatalysts Welded on Nickel Foam for Efficient and Durable Hydrogen Evolution at Ultrahigh Current Densities [J]. Advanced Energy Materials, 2023, 13(1): 2202595.
- [R3.8] Liu T, Vaudin M D, Bunn J R, et al. Quantifying dislocation density in Al-Cu coatings produced by cold spray deposition[J]. Acta Materialia, 2020, 193: 115-124.
- [R3.9] Zhang, Yan, et al. "Dislocation-engineered piezocatalytic water splitting in single-crystal BaTiO₃." Energy & Environmental Science (2025).
- [R3.10] Hao, Shengnan, et al. "Effect of crystal defects on the electrocatalytic CO₂ reduction performance of pure copper." Scripta Materialia 252 (2024): 116268.
- [R3.11] Kim, Hyo-Jin, et al. "Robust Self-Catalytic Reactor for CO₂ Methanation Fabricated by Metal 3D Printing and Selective Electrochemical Dissolution." Advanced Functional Materials 33.41 (2023): 2303994.
- [R3.12] Han C, Cheng C, Lv W, et al. Hydrogen Spillover Enabled by Edge Dislocations for Efficient Hydrogen Evolution[J]. Advanced Functional Materials, 2025: 2425615.
- [R3.13] Thiagarajan S, Thaiyan M, Ganesan R. Physical vapor deposited highly oriented V₂O₅ thin films for electrocatalytic oxidation of hydrazine[J]. RSC advances, 2016, 6(86): 82581-82590.
- [R3.14] Bierman M J, Lau Y K A, Kvit A V, et al. Dislocation-driven nanowire growth and Eshelby twist[J]. Science, 2008, 320(5879): 1060-1063.
- [R3.15] Meng F, Jin S. The solution growth of copper nanowires and nanotubes is driven by screw dislocations[J]. Nano letters, 2012, 12(1): 234-239.
- [R3.16] Ishida Y, Ishida H, Kohra K, et al. Determination of the Burgers vector of a dislocation by weak-beam imaging in a HVEM[J]. Philosophical Magazine A, 1980, 42(4): 453-462.
- [R3.17] Romaner L, Pradhan T, Kholobina A, et al. Theoretical investigation of the 70.5° mixed dislocations in body-centered cubic transition metals[J]. Acta materialia, 2021, 217: 117154.
- [R3.18] Yu, Y., Wang, C., Yu, Y., Wang, Y. & Zhang, B. Promoting selective electroreduction of nitrates to ammonia over electron-deficient Co modulated by rectifying Schottky contacts. Science China Chemistry 63, 1469-1476 (2020).
- [R3.19] Thirathipviwat, P. et al. A comparison study of dislocation density, recrystallization and grain growth among nickel, FeNiCo ternary alloy and FeNiCoCrMn high entropy alloy. Journal of Alloys and Compounds 790, 266-273 (2019).
- [R3.20] Varvenne, C., Luque, A. & Curtin, W. A. Theory of strengthening in fcc high entropy alloys. Acta Materialia 118, 164-176 (2016).
- [R3.21] Gallet J, Perez M, Guillou R, et al. Experimental measurement of dislocation density in metallic materials: A quantitative comparison between measurements techniques (XRD, R-ECCI, HR-EBSD, TEM) [J]. Materials Characterization, 2023, 199: 112842.
- [R3.22] Cui L, Yu C H, Jiang S, et al. A new approach for determining GND and SSD densities based on indentation size effect: An application to additive-manufactured Hastelloy X[J]. Journal of Materials Science & Technology, 2022, 96: 295-307.
- [R3.23] Zhu C, Harrington T, Gray III G T, et al. Dislocation-type evolution in quasi-statically compressed polycrystalline nickel[J]. Acta Materialia, 2018, 155: 104-116.
- [R3.24] Qu, Shuo, et al. "Anisotropic material properties of pure copper with fine-grained

microstructure fabricated by laser powder bed fusion process." *Additive Manufacturing* 59 (2022): 103082.

[R3.25] Hao, Shengnan, et al. "Effect of crystal defects on the electrocatalytic CO₂ reduction performance of pure copper." *Scripta Materialia* 252 (2024): 116268.

[R3.26] Zhao, Jianxiong, et al. "Accurate control of core-shell upconversion nanoparticles through anisotropic strain engineering." *Advanced Functional Materials* 29.44 (2019): 1903295.

[R3.27] Wang, Jia-Qi, et al. "Laser-generated grain boundaries in ruthenium nanoparticles for boosting oxygen evolution reaction." *ACS Catalysis* 10.21 (2020): 12575-12581.

[R3.28] Liu, Siliang, et al. "Extreme environmental thermal shock induced dislocation-rich Pt nanoparticles boosting hydrogen evolution reaction." *Advanced Materials* 34.2 (2022): 2106973.

[R3.29] Zhou, Miao, et al. "Dislocation Network-Boosted PtNi Nanocatalysts Welded on Nickel Foam for Efficient and Durable Hydrogen Evolution at Ultrahigh Current Densities." *Advanced Energy Materials* 13.1 (2023): 2202595.

[R3.30] Wang, Junpeng, et al. "Lattice Strain and Surface Activity of Dislocation-Distorted AgPd Nanoalloys Under Preoxidation and Catalysis Condition." *Small Structures* 4.12 (2023): 2300169.

[R3.31] Fang, Jia-Yi, et al. "Ampere-level current density ammonia electrochemical synthesis using CuCo nanosheets simulating nitrite reductase bifunctional nature." *Nature Communications* 13.1 (2022): 7899.

[R3.32] Wang, Ruhan, et al. "Tuning the Acid Hardness Nature of Cu Catalyst for Selective Nitrate-to-Ammonia Electroreduction." *Angewandte Chemie International Edition*: e202425262.

[R3.33] Dai, Jie, et al. "Spin polarized Fe1-Ti pairs for highly efficient electroreduction nitrate to ammonia." *Nature Communications* 15.1 (2024): 88.

[R3.34] Fang, Jia-Yi, et al. "Ampere-level current density ammonia electrochemical synthesis using CuCo nanosheets simulating nitrite reductase bifunctional nature." *Nature Communications* 13.1 (2022): 7899.

[R3.35] Gan, G.; Hong, G.; Zhang, W. Active Hydrogen for Electrochemical Ammonia Synthesis. *Adv. Funct. Mater.* 2024, 2,2401472.

[R3.36] Cheng, Tao, Hai Xiao, and William A. Goddard III. "Full atomistic reaction mechanism with kinetics for CO reduction on Cu (100) from ab initio molecular dynamics free-energy calculations at 298 K." *Proceedings of the National Academy of Sciences* 114.8 (2017): 1795-1800.

[R3.37] Zheng, Shisheng, et al. "The loss of interfacial water-adsorbate hydrogen bond connectivity position surface-active hydrogen as a crucial intermediate to enhance nitrate reduction reaction." *Journal of the American Chemical Society* 146.39 (2024): 26965-26974.

Response to Reviewers

Harnessing screw dislocations in shell-lattice metamaterials for efficient, stable electrocatalysts

Liqiang Wang[†], Di Yin[†], James Utama Surjadi[†], Junhao Ding, Huangliu Fu, Xin Zhou, Rui Li, Mengxue Chen, Xinxin Li, Xu Song^{*}, Johnny C. Ho^{*}, Yang Lu^{*}

We sincerely appreciate for the comprehensive comments and constructive suggestions raised by reviewers. Based on that, we have revised our manuscript accordingly. We believe the manuscript has been improved. In the following context, we have prepared a response to the comments in a point-by-point fashion, detailed below (red: comments from the reviewer; black: response; blue: revised manuscript)

Reviewer #1

General comment:

The authors have addressed the concerns well.

Response: Thank you for your positive feedback and for acknowledging our efforts to address the concerns raised.

Reviewer #2

General comment:

I sincerely appreciate the authors' efforts in addressing all my comments effectively except one minor unresolved question regarding energy efficiency in comment 2.5.

Response: We appreciate your acknowledgment of the improvements made in addressing most of your comments. Regarding the unresolved question about energy efficiency, we have added the comment 2.1 below to explicitly address this concern.

Comment 2.1

Energy efficiency (EE) for an electrocatalytic process can be calculated using the applied potential and Faradaic efficiency. It is defined as the percentage of energy reserved in the targeted product (here ammonia) divided by the total energy input to the electrolysis system. Could the authors provide the calculated EE value based on either Figure S21 or 6f.

Additionally, in the three-chamber reactor, the authors mentioned Nafion 117 membrane permitted the transport of protons from the anode to the cathode to counteract the OH⁻ produced at the cathode and

maintain a constant pH in the cathodic electrolyte. While Nafion's high proton conductivity (up to 0.2 S/cm) allows for proton transport between compartments, I'm concerned that this transport may not be sufficient to fully equilibrate the pH between the two sides during electrolysis under the applied reaction rate. Could the authors clarify this.

Response: We sincerely thank the reviewer for pointing out the energy efficiency calculation and the potential pH variations during electrolysis.

a) Calculation of Energy Efficiency

Energy efficiency (EE) is a critical metric for evaluating the viability of electrocatalytic processes, particularly for ammonia synthesis. Below, we provide a comprehensive explanation of the EE calculation methodology. Our analysis is based on the experimental data presented in **Figure S21**.

The half-cell energy efficiency was defined as the ratio of fuel energy to applied electrical power, which was calculated by:

$$EE_{NH_3} = \frac{(E_{OER}^{\theta} - E_{NH_3}^{\theta}) \times FE_{NH_3}}{E_{OER} - E_{NH_3}}$$

E_{OER}^{θ} is the equilibrium potential of OER (1.23 V versus RHE). $E_{NH_3}^{\theta}$ represents the equilibrium potential of nitrate electroreduction to ammonia, which is reported as $E_{NH_3}^{\theta} = 0.69$ V vs. RHE under alkaline conditions (pH 14) (*Journal of the American Chemical Society* 142.12 (2020): 5702-5708). The Pourbaix diagram for ammonia provides the equilibrium potentials at different pH values (*Advanced Functional Materials* 31.11 (2021): 2008533, *Nature Catalysis* 6.12 (2023): 1125-1130, *Small Methods* 4.12 (2020): 2000672). FE_{NH_3} is the Faradaic efficiency for NH_3 . E_{OER} is 1.23 V versus RHE (assuming the overpotential of the water oxidation is zero), and E_{NH_3} is the applied potentials in NO_3^- -to- NH_3 electroreduction (*Nature Communications* 15.1 (2024): 8583, *Nature Catalysis* 6.5 (2023): 402-414).

We calculated EE for FeCoNi-S4650 and FeCoNi-LBL4650 at various applied potentials. The analysis confirms that FeCoNi-S4650 achieved higher EE than FeCoNi-LBL4650 across a broad potential window of -0.1 to -1.1 V. Notably, FeCoNi-S4650 exhibited a peak NH_3 cathodic half-cell EE of 30.3% at an applied potential of -0.5 V (**Figure S21 b**).

We added the calculation result of EE in revised **Figure S21** and revised related sentence, as shown below. The calculation methodology for energy efficiency has been systematically expounded in *Supplementary Method 4*:

“We also calculated energy efficiency (EE) for FeCoNi-S4650 and FeCoNi-LBL4650 at various applied potentials. The analysis confirms that FeCoNi-S4650 achieved higher EE than FeCoNi-LBL4650 across a broad potential window of -0.1 to -1.1 V. Notably, FeCoNi-S4650 exhibited a peak NH_3 cathodic half-cell EE of 30.3% at an applied potential of -0.5 V (Fig. S21 b).”

Figure. S21 FE_{NH_3} (a) and EE_{NH_3} (b) of FeCoNi-S4650 and FeCoNi-LBL4650 at varied initial potentials. Time-dependent pH variation of FeCoNi-S4650 (c) and FeCoNi-LBL4650 (d) at varied potentials.

b) Investigation of pH Variations

To analyze proton transport through Nafion membranes, we begin by studying the generation of protons at the anode and hydroxides at the cathode.

The anodic half-cell reaction is shown below:

The cathodic half-cell reaction is shown below:

By balancing all the charges and ions involved in the aforementioned reactions (assuming complete migration of all H^+ ions generated from the anodic OER to the cathode to neutralize the OH^- groups generated during NO_3RR under idealized conditions), the net full-cell reaction can be presented as (*Nature Catalysis* 7.9 (2024): 1032-1043):

Even under idealized assumptions of complete H^+ migration from anode to cathode, the net full-cell reaction inherently generates residual hydroxide ions, fundamentally dictating cathodic pH elevation. Crucially, the practical H^+ diffusion flux is coordinated constrained by multiple factors (such as membrane conductivity, Butler-Volmer kinetics at interfaces, and mass transport limitations), resulting in the persistent challenge of incomplete H^+ migration. Thus, this dual limitation, stoichiometrically mandated OH^- accumulation and kinetically restricted proton delivery, ensures unavoidable pH increases at the cathode during practical conditions (*Chinese Journal of Catalysis* 53 (2023): 8-12., *Advanced Energy Materials* 12.44 (2022): 2202247).

To validate the theoretical predictions, we conducted experimental measurements to monitor dynamic pH variations within the cathode electrolyte during operation. pH monitoring (Figure S21 c-d) for FeCoNi-S4650 and FeCoNi-LBL4650 catalyst showed catholyte pH rises within 1 hour at our applied potential, confirming that proton transport was insufficient to prevent excessive alkalization. That is, dynamic electrolysis establishes a pH gradient rather than full equilibration. The membrane allows partial H^+/OH^- neutralization, but continuous OH^- production at the cathode maintains a residual alkalinity (*Nature Materials* (2025): 1-8).

Overall, both theoretical analysis and experimental characterizations indeed demonstrate that pH increases in cathode during nitrate reduction process. Therefore, we removed the inaccurate description in the previous manuscript and added the description of pH variations in NO_3RR in the revised manuscript.

“pH monitoring (Figure S21 c-d) for FeCoNi-S4650 and FeCoNi-LBL4650 catalyst showed catholyte pH rises within 1 hour at our applied potential, confirming that proton transport was insufficient to prevent excessive alkalization.”

Reviewer #3

General comment:

The authors have made visible efforts to address the comments raised by the reviewers, and the revised manuscript reflects some improvements in clarity and organization. Their responses indicate a attempt to clarify the modeling framework and experimental validation, which is appreciated.

However, despite these revisions, I find that the overall depth, and scientific rigor of the work still fall short of the standards expected for publication in this journal. The core contributions remain somewhat incremental, and several aspects of the modeling assumptions, data interpretation, and novelty of the findings are not convincingly developed. In light of these considerations, I believe that the manuscript, in its current form, would be more appropriately directed toward a more specialized or lower-tier journal. I therefore do not recommend publication in this journal.

Response: We sincerely appreciate the reviewer’s recognition of our efforts to improve clarity and transparency. We would like to address the concerns directly by summarizing the key scientific contributions and novelty, and rigor of our work.

Two Key Innovations:

- **Scalable 3D Printing with Nanoscale Feature:**

We introduce a **scalable additive manufacturing strategy** that incorporates screw dislocations under low supersaturation conditions to achieve hierarchical structures spanning from nanometers to centimeters. This overcomes the resolution limitations of existing 3D printing technologies, which thus far have not been able to produce nanoscale features essential for high-performance electrocatalysis (**Figure R1**, *Joule*, 2019, 3(8): 1835-1849; *Nature Reviews Chemistry*, 2019, 3(5): 305-314). The bulk scale 10^7 times larger than their smallest nanoscale feature sizes within the metamaterials, making it outstanding candidate for high-performance catalytic electrodes.

Accordingly, we added the following sentence to further highlight the manufacturing advantage of our 3D printing strategy in the manuscript:

“The metamaterials exhibit a bulk scale 10^7 times larger than their smallest nanoscale feature sizes within the metamaterials, establishing them as exceptional candidates for high-performance catalytic electrodes.”

- **Programmable Curvature for 3D Lattice Strain Engineering:**

Curved substrates have become a prominent strategy for generating lattice strain, enhancing catalytic performance through tuning the electronic configuration of metal sites. (*Nature Reviews Materials* 2.11 (2017): 1-13.). However, most substrate topographies are passively generated and exhibit significant randomness, fundamentally limiting the controllability of lattice strain.

We demonstrate that 3D-printed curvature control can directly modulate surface lattice strain due to the simultaneous synthesis of curved substrates and surface nanostructures ensures strong metallic bonding and maximizes elastic strain energy accumulation during the growth process. This overcomes the randomness and weak interface bonding of previous approaches and introduces a novel method for **3D strain engineering** in catalysis (*Advanced Materials*, 2023, 35(39), 2209876).

We believe that these two advances collectively bridge a critical gap between **macroscale catalytic electrode manufacturing** and **nanostructure design**, offering a new paradigm for integrated, scalable, and tunable electrocatalyst fabrication.

Figure R1. The resolution gap between the current 3D printing methods and the need for high-performance catalytic electrodes.

Scientific Depth and Rigor:

Our work combines technology development, mechanistic insight, and application validation (**Figure R2**):

- **Mechanistic Insight:**
Unlike conventional morphology-focused nanomaterial catalyst studies (*Nature Catalysis*, 2023, 6(5): 402-414; *Nature Communications*, 2022, 13(1): 1129; *Nature Synthesis*, 2023, 2(7): 624-634), our work delivers deeper mechanistic insights into synthesis pathways and dislocation multiplication through advanced multiscale characterization tools, such as EBSD, TEM, STEM, and APT.
- **Defect-Strain Coupling Framework:**
Existing studies often fail to deeply discuss the origins of lattice strain and its intricate relationships with growth behavior (*Nature*, 2021, 598(7879): 76-81; *Nature Energy*, 2019, 4(2): 115-122; *Nature Communications*, 2022, 13(1): 4200). We establish a fundamental relationship between curvature-induced elastic energy, dislocation density, and resulting lattice strain, which is an area often underexplored in prior work.

- **Catalytic Performance + DFT Modeling:**
Our DFT atomistic modeling framework integrates dislocation configurations and strain magnitudes quantitatively validated by correlative HRTEM/XRD analyses. In response to Reviewer #2's and #3's previous concerns, we systematically expanded our modeling to include edge dislocations, establishing a multi-scale connection between defect structures and their catalytic impact.
- **Application Relevance:**
We demonstrate device-level integration through a custom-designed three-chamber reactor to continuously produce and capture ammonia products, further highlighting the industrial application potential.

Figure R2. Research framework diagram.

In summary, we respectfully believe that our manuscript presents two key innovations:

- A **novel, scalable 3D printing framework** for hierarchical catalytic electrodes;
- A **new mechanism for 3D lattice strain engineering** via metamaterial curvature control.

Regarding scientific depth and rigor, we conducted:

- A rigorous, multiscale investigation of growth, defects, strain, and performance;
- A viable path validation to practical implementation via integrated device testing.

We hope this response clarifies our contributions and addresses your concerns regarding novelty, rigor, and impact, meeting the requirements for *Nature Communications*.